# Transcriptional heterogeneity shapes stress-adaptive responses in yeast

Mariona Nadal-Ribelles [1,2] ✉, Guillaume Lieb [3], Carme Solé [1,2], Yaima Matas [1,2], Ugo Szachnowski [4], Sara Andjus [4], Maria Quintana [1,2], Mònica Romo [1,2], Aitor Gonzalez Herrero [1,2], Antonin Morillon [4], Serge Pelet [3], Eulàlia de Nadal [1,2] ✉ & Francesc Posas [1,2] ✉

In response to stress, cells activate signaling pathways that coordinate broad changes in gene expression to enhance cell survival. Remarkably, complex variations in gene expression occur even in isogenic populations and in response to similar signaling inputs. However, the molecular mechanisms underlying this variability and their influence on adaptive cell fate decisions are not fully understood. Here, we use scRNA-seq to longitudinally assess transcriptional dynamics during osmoadaptation in yeast. Our findings reveal highly heterogeneous expression of the osmoresponsive program, which organizes into combinatorial patterns that generate distinct cellular programs. The induction of these programs is favored by global transcriptome repression upon stress. Cells displaying basal expression of the osmoresponsive program are hyper-responsive and resistant to stress. Through a transcription-focused analysis of more than 300 RNA-barcoded deletion mutants, we identify genetic factors that shape the heterogeneity of the osmostress-induced transcriptome, define regulators of stress-related subpopulations and find a link between transcriptional heterogeneity and increased cell fitness. Our findings provide a regulatory map of the complex transcriptional phenotypes underlying osmoadaptation in yeast and highlight the importance of transcriptional heterogeneity in generating distinct adaptive strategies.

Cell adaptation to stress requires the modulation of various aspects of cell physiology, including gene expression[1]. In eukaryotes, stress-activated protein kinases (SAPKs) receive and transmit extracellular cues to the intracellular environment, leading to the modulation of cellular functions, such as cell cycle arrest and shifts in metabolic fluxes to promote adaptation. One of the main outcomes of SAPK activation is the tight regulation of gene expression, which, in response to stress, dynamically induces the up- and down-regulation of a large gene set, affecting 20% of the transcriptome[2,3]. In *Saccharomyces cerevisiae* (*S. cerevisiae*), Hog1 SAPK is phosphorylated in seconds in response to stress and rapidly accumulates in the nucleus, where it directly associates with target genes, an event that ultimately induces the expression of a set of osmoresponsive genes[4–6]. Osmoresponsive genes include the core environmental stress-responsive genes (iESR), a set of genes induced by virtually all stressors, together with osmostress-specific genes, such as those related to glycerol import and biosynthesis[3]. This transient upregulation of gene expression is accompanied by the active downregulation of proliferation-related genes (rESR), including ribosomal genes (RiBi), and a global decrease in gene expression and mRNA stability[3,7]. Hog1 bypasses the global

[1]Department of Medicine and Life Sciences, Universitat Pompeu Fabra, Barcelona 08003, Spain. [2]Institute for Research in Biomedicine (IRB Barcelona), The Barcelona Institute of Science and Technology, Barcelona 08028, Spain. [3]Department of Fundamental Microbiology, Faculty of Biology and Medicine, University of Lausanne, Lausanne, Switzerland. [4]ncRNA, Epigenetic and Genome Fluidity, Institut Curie, Sorbonne Université, CNRS UMR3244, F-75248, Paris Cedex 05, France. ✉e-mail: mariona.nadal@irbbarcelona.org; eulalia.nadal@upf.edu; francesc.posas@irbbarcelona.org

transcriptional repressive state by directly regulating transcription initiation by binding to target genes through transcription factors such as Msn2/4, Sko1 and Hot1, among others, to then recruit chromatin-modifying enzymes and transcriptional machinery (RNA Pol II and associated factors)[5,6,8]. Additionally, Hog1 serves as a selective elongation factor by traveling with RNA Pol II and associating with chromatin remodelers through the target coding regions[4,9–11]. Once cells have adapted, Hog1 is dephosphorylated and re-shuttled to the cytoplasm, and homeostatic transcription is restored.

Bulk biochemical studies have highlighted the key players involved in osmostress-induced transcriptional reprogramming. Conversely, single-cell analyses have uncovered a paradigmatic scenario in which the transcriptional output of a gene can differ dramatically from others, despite robust and identical Hog1 signaling[12]. Single gene reporters and single-molecule fluorescence in situ hybridization (smFISH) measurements have revealed that impaired chromatin remodeling strongly contributes to the generation of bimodal distributions of transcription at stress-induced loci, thereby pointing to additional regulation downstream of signaling inputs[12–15]. In parallel, the expression noise of stress-responsive genes has been described through the oscillatory behavior of the Msn2 transcription factor, and the basal expression of some stress-responsive genes has been detected by scRNA-seq across several studies and observed by time-lapse microscopy, suggesting residual expression of stress-responsive genes within a population[16–18]. However, the effect of transcriptional heterogeneity on stress adaptation has not yet been addressed in depth. In recent years, we have seen the successful implementation of several scRNA-seq protocols suitable for yeast[19], but longitudinal profiling of dynamic environments has not been explored to date[20]. Furthermore, multiplexed integration with genetic and environmental perturbations has been restricted to a limited number of mutants because of the lack of global barcoding strategies, such as those generated by CRISPR-gRNAs (Perturb-seq), a widely used system in mammalian cells[21,22]. In this study, we applied longitudinal scRNA-seq profiling, together with comprehensive genetic perturbations targeting transcriptional components, to highlight the role of transcriptional heterogeneity in generating distinct subpopulations of cells with different adaptive potential.

## Results

### Osmoadaptation increases transcriptional heterogeneity

Most of our understanding of transcriptional adaptive responses is derived from bulk assays. However, evidence from single-cell transcriptional readouts in response to stress suggests a great degree of heterogeneity within a population. We systematically characterized stress responses by generating a time-resolved, single-cell transcriptional map using wild-type (WT) and hog1 mutant strains as references for full and impaired transcriptional responses. We cultured wild-type and hog1 cells carrying unique marker combinations and performed longitudinal scRNA-seq in response to osmostress (Fig. 1a). We profiled over 21,000 cells and applied stringent filtering criteria, removing low-quality cells (<500 or >3000 genes) and those expressing more than one marker (Supplementary Fig. 1a). Overall, we retained 19,866 (93.5%) high-quality singlets with assigned genotypes for further analysis. Each genotype and condition were represented by more than 1500 cells, with a mean of 1290 genes and 3009 molecules detected per cell (Supplementary Fig. 1b).

To generate an unbiased overview of the transcriptional changes caused by stress, we regressed cell cycle effects and excluded ribosomal genes before performing unsupervised principal component analysis (PCA) (see Methods). Under control conditions, WT and hog1 cells showed overlapping clusters, thereby indicating similar transcriptional profiles. In contrast, after 5 minutes of osmostress, WT cells formed a distinct, more heterogeneous cluster that was most pronounced at the peak of the response (15 min) and became less distinct at a later time point (30 min) (Fig. 1b, and Supplementary Fig. 1c). At short time points hog1 overlapped with control and remained closer in the PCA across times. These findings thus highlight the role of the Hog1 SAPK in the induction of the adaptation program. As expected, we found stress-responsive genes as the main drivers of clustering (Supplementary Fig. 1d), pointing out these genes as a major source of intra-genotype heterogeneity.

To assess the dynamics of the stress-responsive genes, we tracked the expression pattern of 200 consistently induced and repressed osmoresponsive genes obtained from 5 independent bulk RNA-seq experiments (hereafter referred to induced/repressed osmoconsensus) (Supplementary Data 1)[23]. As expected, under control conditions, the expression of the osmoconsensus program across most cells was very low in both WT and hog1 genotypes. However, in response to stress, there was a rapid and transient induction of the transcriptional response, which peaked at 15 min and was strongly impaired in hog1 cells (Fig. 1c). Repressed genes also revealed a stronger response in WT compared to hog1 cells, consistent with bulk analyses of these signatures (Fig. 1d). The increase in the variability of gene expression, particularly for the induced programs, was the highest at the peak of the response while unresponsive genes remained constant (Fig. 1e, Supplementary Fig. 1e–g). This increased heterogeneity extends beyond individual genes to entire transcriptional programs, confirming observations from single-gene stress-responsive reporters[12]. Thus, the single-cell resolution revealed that stress induced a high level of transcriptional variability, particularly within the induced osmoconsensus genes.

### Cells display heterogeneous use of the osmoresponsive program in response to stress

A potential source of the cell variability in stress response could be differential usage of the induced genes. For each gene in the osmoconsensus signature, we calculated the percentage of cells that expressed a particular transcript and their average expression (Fig. 2a). Under control conditions, few genes were expressed in both WT and hog1 mutant cells. After 5 minutes of stress (0.4 M NaCl), the percentage of cells expressing stress-responsive genes increased, but with low transcript levels. At 15 min, expression peaked, with an increase in the average expression of osmoconsensus genes. The percentage of cells expressing was similar in both strains, but the strength of the response was diminished in hog1 mutant (Fig. 2a). This observation suggests that gene usage is inherent to the transcription unit as the percentage of expressing cells showing a degree of similarity between WT and hog1 mutant cells, whereas transcriptional output was strongly regulated by the SAPK.

Remarkably, analysis of the expression of the whole osmoconsensus program at the peak of expression in WT cells (15 min of stress) revealed that only a few genes (less than 25%) were expressed in most of the cells (>75%), whereas the rest of the genes were expressed in only a fraction of the population (Fig. 2b, Supplementary Fig. 2a, and Supplementary Data 2). Similar results were observed for repressed genes. In this regard, hog1 mutant cells failed to induce and repress gene expression (Supplementary Fig. 2b, c). Thus, cell-to-cell variability in the stress response can be attributed to a combinatorial expression pattern with only few genes shared by the majority of the population.

We hypothesized that differential gene expression creates a cell-specific osmoresponsive fingerprint. By scoring expressed genes per cell at peak expression (WT 15 min, avg. expression >0), we found a positive correlation between detected genes and average osmoconsensus program expression (Fig. 2c). The wild-type population expressed 93 out of 200 genes (46.5%) (Fig. 2d), with this pattern reproduced by at least 52% of cells. While top-expressing genes were consistent, selective gene induction generated unique expression profiles across individual cells (Fig. 2e). Our findings reveal that bulk

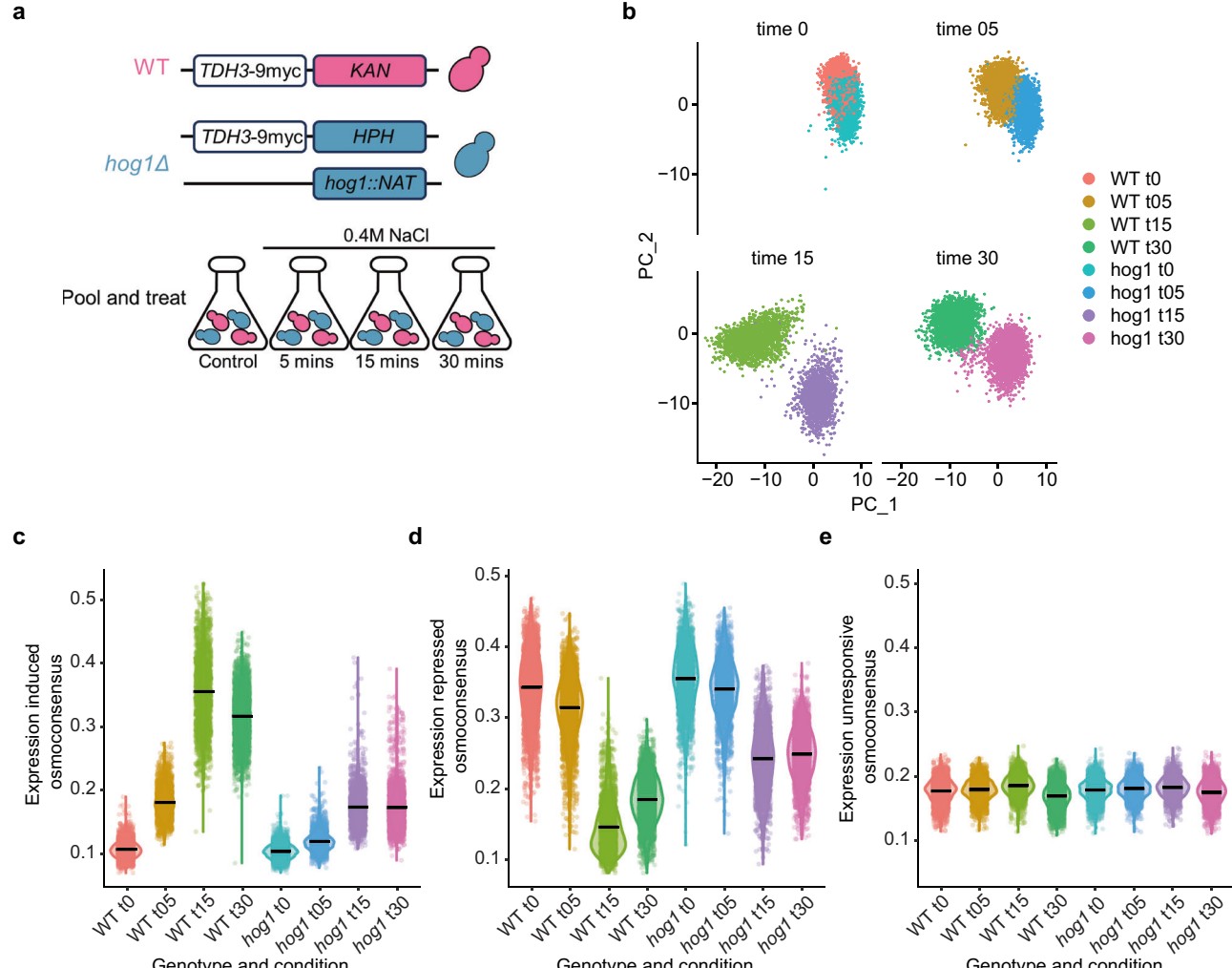

**Fig. 1 | Osmoadaptation increases transcriptional heterogeneity. a** Schematic representation of the experimental design. Wild-type cells (BY4741) and derivative hog1 mutant cells carrying distinct selectable markers. Wild type cells carry an endogenous tagged *TDH3−9myc::KAN* and hog1 mutant cells (*hog1::NAT*) carry and endogenously tagged *TDH3−*9myc::HPH. Expression of the KAN, NAT and HPH markers are under a constitutive promoter. Cells were grown individually and mixed at a 1:1 ratio. Pooled cells were then harvested in control conditions (t0) or subjected to osmostress for 5, 15 or 30 min by treatment with 0.4 M NaCl, and a single 10X v3.1 library was generated for each time point. **b** Principal component Analysis of each genotype split by the indicated time points. **c**−**e** Distribution of signature expression across the indicated strains and time points for upregulated, repressed, and unresponsive genes (*y*-axis) for the indicated samples. Signature genes were extracted from 5 independent bulk RNA-seq studies[23]. Black bars indicate the mean expression.

---

study observations of osmoresponsive-gene programs arise from heterogeneous gene usage and expression within the population.

## Modular activation of functional genes in response to stress

To assess the regulatory logic of osmoresponsive program heterogeneity, we clustered cells based on differentially expressed genes between control and NaCl-treated cells (15 min). We identified five expression pattern subtypes (Fig. 2f, g). Cluster 0 (33% of the population) showed weak but relatively homogeneous gene expression. Cluster 1 (22% of the population) displayed reduced molecule count, low stress-gene expression, and strong induction of only two genes (*PYC1* and *ADE2*). Cluster 2 exhibited strong modular expression of 29 genes, particularly protein folding-related genes, including osmostress-induced chaperones (*HSP82*, *SSA4*) and the heat shock-specific chaperone *HSC82*[23]. Clusters 3 and 4 shared metabolic and oxidative stress genes (86% gene overlap) (Supplementary Data 3, and Supplementary Fig. 2d). Cluster 4 notably expressed high levels of neighboring osmoresponsive genes *PAI3* and *SPI18*, with the latter encoding a hydrophilic protein inhibiting apoptosis through anti-oxidant effects crucial for desiccation stress (Fig. 2f)[24]. When scoring

the osmoconsensus program, clusters 3 and 4 showed the highest expression of the signature, while cluster 1 scored lowest (Fig. 2g). This differential cluster usage reveals modular gene subprogram expression, suggesting diverse cellular adaptation strategies.

We then reasoned that the co-expression of specific gene sets might reflect different transcription factor activity or selection. To address this question, we generated the signatures of 12 transcription factors involved in the stress response. These included major transcription factors ranging from the general ESR regulators Msn2/4 and the stress-specific Sko1 to more specialized factors such as Hot1 and Mot3, which regulate a small subset of genes[6]. The activity of Msn2/4 was significantly higher in clusters 3 and 4, which showed the greatest variability among transcription factors (Supplementary Fig. 2e). In contrast, most transcription factors like Sko1 and Smp1 showed less variability (7 out of 12 transcription factors displayed this behavior), possibly due to their restricted specificity towards osmoresponsive genes (Supplementary Fig. 2e). However, the activity of specialized factors such as Mot3 differed slightly between clusters 3 and 4, while cells in cluster 2 showed higher activity of Hsf1, consistent with a subset of cells generating a heat stress-like response (Supplementary

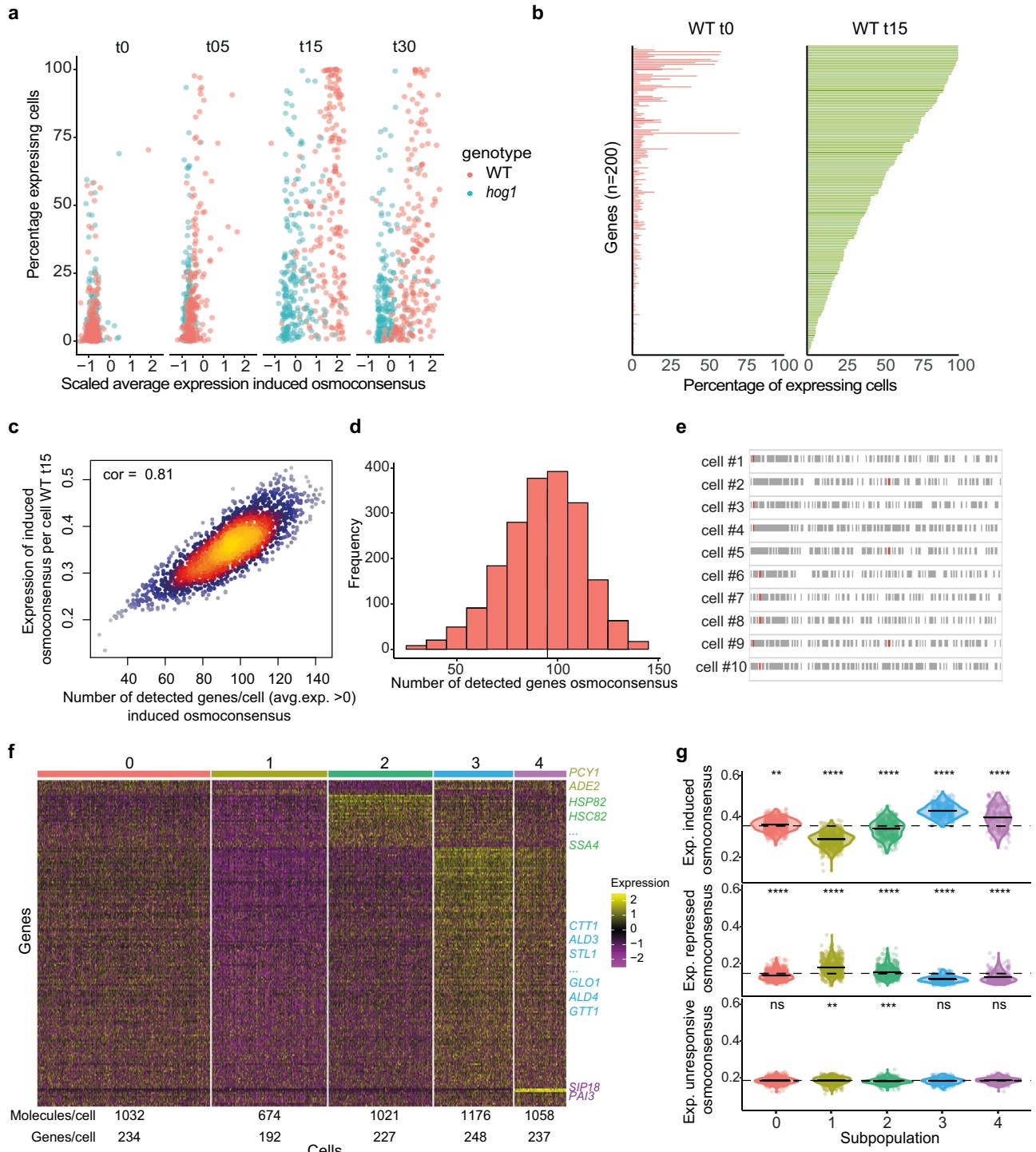

Fig. 2e). Thus, distinct usage of transcription factor activity/selection within the population generates the differential gene programs within the stress response.

## Transcriptional heterogeneity shapes the adaptive capacity of the cells

Leveraging on the unbiased coverage of scRNA-seq, we next examined whether a global transcriptional context influences the strength and topology of the stress response. To this end, we re-clustered the WT 15 minutes dataset using highly variable genes with Louvain clustering. This generated 9 subpopulations (Fig. 3a), each of which characterized by the expression of specific genes (Fig. 3b) and enriched in the corresponding subpopulation (Fig. 3c). When we scored the expression of the induced osmoconsensus, we detected distinct responses to stress depending on the subpopulation: high responders (50% cells, clusters 0, 2, 4 and 7), low responders (32% cells; clusters 1, 5 and 8), or average responders (clusters 3 and 6) (Fig. 3d). To understand the nature of these subpopulations, we performed a Gene Ontology (GO) analysis, considering the cluster-specific upregulated markers. Most of the subpopulations contained genes associated with known physiological processes (Supplementary Fig. 3a and Supplementary Data 4). Cluster 8 exhibited markers of aged cells, expressing known age-related genes like *FIT3* and iron-regulon genes[25,26], with reduced expression of the osmoconsensus signature. Cluster 5 showed

**Fig. 2 | Cells show a heterogeneous use of the osmoresponsive program in response to stress. a** Scatter plot represents the scaled average expression of each of the induced osmoconsensus program ($n = 200$ genes, x-axis) in WT (red points) and *hog1* cells (blue points) against the percentage of expressing cells (y-axis) for the indicated times. **b** Bar plot represents the percentage of cells expressing the induced osmoconsensus signature in control conditions (red bars) or after a 15-min treatment with 0.4 M NaCl in wild-type cells. **c** Correlation of the number of induced osmoconsensus genes per cell (x-axis) and the average induction per cell in the wild-type cells after 15 min of treatment with 0.4 M NaCl. Detected genes are considered if expression/cell >0. Points are colored by density (warmer higher density). Spearman correlation is shown. **d** Histogram of the number of genes as in (**c**). Dashed line indicates the average (93 genes/cell). **e** Binary heatmap of the induced osmoconsensus gene expression footprint (x-axis) per cell (rows). Detected genes (average expression >0) are shown in gray and non-detected genes (average expression =0) in white. For each cell, the gene with the highest expression is highlighted in red. **f** Heatmap of marker genes for subpopulations identified in the WT 15-min dataset. Differential expressed genes (fold-change ≥1.5, adjusted *p*-value < 0.05) obtained by comparing the control and 15-min conditions were used to identify subpopulations using for Louvain clustering. Yellow colors indicate higher whereas purple colors indicate expression levels. Subpopulation labels are shown at the top of the heatmap and representative genes and cluster-specific gene names are shown as well as the median number of molecules and genes per cell. **g** Expression distribution of the induced (top), repressed (middle) and unresponsive (bottom) signatures across the identified subpopulations in (**g**). Dotted line indicates the median expression of the entire population as a reference. Two-sided Wilcoxon test Benjamini–Hochberg adjusted of each cluster against the population is shown above. Symbols ns/*/**/***/**** represent *p*-values > 0.05, <0.05, <0.01, <0.001, <0.0001. Source data are provided as a Source Data file.

deficient response and enrichment in ribosome biogenesis. In contrast, clusters 4 and 7 displayed above-average osmoconsensus signature expression and enrichment in daughter-specific genes (DSE). Indeed, daughter cells systematically displayed higher expression of the osmoconsensus signature throughout the time course examined (Supplementary Fig. 3b). This unbiased clustering recapitulated the heat stress-like response (clusters 3 and 4, Supplementary Fig. 3a) and the metabolism/oxidative stress signatures (cluster 0, Supplementary Fig. 3a) that were identified in the aforementioned clustering performed on stress-responsive genes (Fig. 2f), hence validating that they represent two distinct subpopulations with defined transcription modes.

Of note, cells in cluster 0 comprised the subpopulation with the highest expression of the osmoconsensus program (Fig. 3d). This finding could be explained by either the higher expression of globally used genes and/or the co-expression of a large number of genes of the osmoconsensus program. Thus, we first tested the activity of twelve transcription factors involved in osmoadaptation. Cluster 0 consistently scored as the top expressers of the main transcription factors (Msn2/4, Sko1 and Hot1), except for Smp1 and Hsf1 (Fig. 3e). Given the wider transcription factor activity, we assessed whether these cells could co-express a larger fraction of genes. We calculated the co-expression degree of five representative genes with distinct expression frequencies ranging from 100% to 10% (see Methods). This comparison identified a subset of hyper-responsive cells ($n = 180$) with a higher degree of co-expression, 156 cells (86%) of which belonged to cluster 0 (Fig. 3f). Hyper-responsive cells showed broader gene use with very mild changes in maximal gene expression not attributable to the number of molecules/cell (Fig. 3g, h and Supplementary Fig. 3c–e). Therefore, there is a subpopulation of hyper-responsive genes in cluster 0 that are likely to show greater fitness in response to stress.

To validate the phenotype of hyper-responsive cells, we integrated an unstable fluorescent reporter using the *HXT5* reporter and terminator (p*HXT5*-UbiM-mCherry-t*HXT5*). We then isolated top *HXT5*-expressing cells by Fluorescent Activated Cell Sorting (FACS) and assayed them in a competition assay (Supplementary Fig. 3f). Hyper-responsive cells showed greater competitive fitness when compared to a WT strain under stress conditions (Fig. 3i). Overall, our data indicate that the global transcriptional state influences gene selection and total transcriptional output, generating subpopulations with differential stress resistance phenotypes.

**Stochastic expression of stress programs in the absence of stress**
Despite the typically repressed and uniform osmoconsensus signature under control conditions, a small subset of wild-type and hog1 mutant cells unexpectedly showed osmoresponsive gene expression levels comparable to early stress responses. (WT 5 min, Fig. 1c). We clustered WT and *hog1* mutant cells under control conditions individually and projected them onto their respective UMAPs (Fig. 4a, b

and Supplementary Fig. 4a, b). Almost 6% of WT cells showed higher expression of the osmoresponsive genes when compared to the overall population. WT cells did not form a distinct cluster, although there was a biased distribution towards clusters 3, 6 and 7, which were enriched in daughter cells (DSE for clusters 3 and 6) (Fig. 4c, Supplementary Fig. 4a and Supplementary Data 5). Surprisingly, this basal expression was Hog1-independent, since 3% of *hog1* mutant cells showed similar basal expression (Supplementary Fig. 4b, c). This finding suggests that the basal firing of the stress-responsive program is partially independent of the upstream signaling/activation of the HOG pathway, which is required for the transient induction of stress-responsive genes.

To identify which genes define basal-stressed cells under basal conditions, we performed differential expression analysis comparing basal-stress cells (top 10% of cells) to the rest of the WT population (Supplementary Fig. 4d, and Supplementary Data 6). We found that the basally stressed population comprised cells that displayed low expression of histone genes and induced the expression of several stress−responsive genes (Fig. 4d), mostly regulated by both transcription factors Msn2/4 and Sko1 among others (Supplementary Fig. 4e). Basal stress-gene expression is lower in S phase cells but reflects a weak but general activation of the main program rather than leaky expression of a single gene. To assess the impact of the basal expression of the stress program, we used two strategies, namely monitoring nascent transcription through microscopy and assessing expression using destabilized fluorescent reporters fused to gene promoters via flow cytometry. First, to assess nascent transcripts, we implemented the PP7/MS2 system to trace the production of mRNA by microscopy, as reported before for osmoresponsive genes[25]. For each gene of interest, we integrated a reporter that consisted of the promoter of the gene of interest followed by an array of 24 MS2/PP7 loops in a WT strain constitutively expressing fluorescently tagged viral protein MS2/PP7. Promoter activation led to fluorescent foci accumulation at the transcription site due to binding of nascent transcripts with fluorescent phage coat protein. Second, we generated destabilized fluorescent reporters whose expression is controlled by the promoter and terminator of the gene of interest, and we measured fluorescence as a proxy of transcriptional output using FACS.

We followed basal stressed cells over time in WT cells expressing the p*HOR7* reporter gene using the MS2-GFP system (maker for basal stressed cells) and also the p*HSP12*-PP7-mCherry (an osmoresponsive marker gene). We simultaneously assessed the transcriptional output of p*HOR7* and p*HSP12* in single cells before and after stress and measure p*HSP12* expression as a function of basal p*HOR7* activity. High *HOR7* expressing cells also showed elevated *HSP12* expression, suggesting that basal stressed cells have a stronger response to stress ($pval = 4.0201e{-}11$) (Fig. 4e, Supplementary Fig. 4f). We hypothesized that a faster transcriptional response would provide a selective advantage in response to stress. Thus, we created a destabilized fluorescent reporter controlled by the *HOR7* promoter and terminator

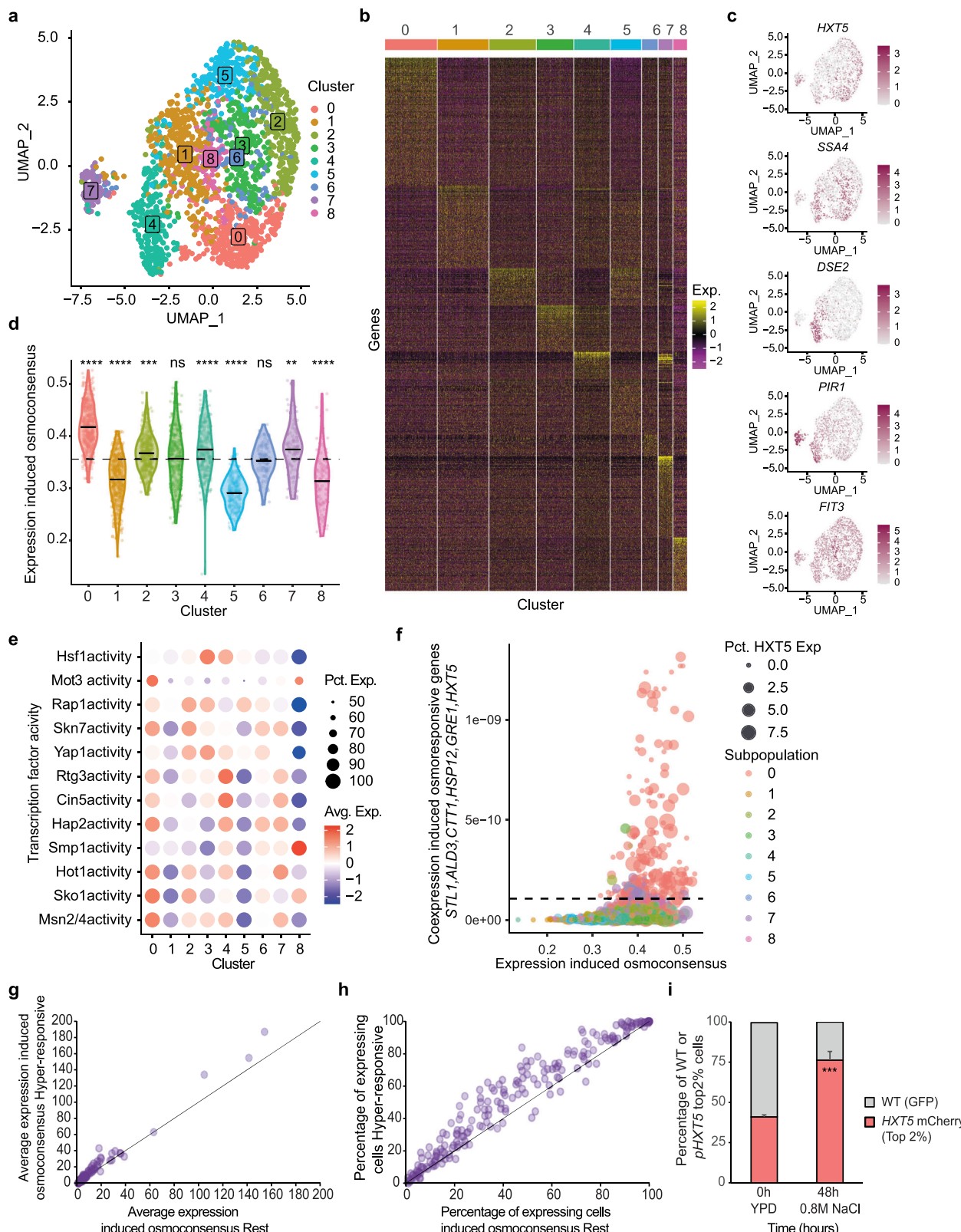

(mCherry). The top 2% of HOR7-expressing cells, isolated by FACS, showed lower fitness under normal conditions but higher fitness during osmostress when compared to WT cells (Fig. 4f). These results suggest that basal expression provides a bet-hedging strategy and that pre-existing basal stressed ("adapted cells") cells can contribute to the appearance of hyper-responsive cells with greater adaptive potential (persister cells).

## Global transcriptome repression favors the induction of the osmoresponsive program

Stress-inducible gene expression coincides with repression of certain genes and overall genome-wide downregulation[1,3,7,26]. Bulk studies suggest that efficient stress-responsive gene induction may depend on cells ability to redistribute RNA Pol II to these genes[5,8]. We harnessed the single-cell resolution to examine the

**Fig. 3 | Transcriptional heterogeneity shapes the adaptive capacity of the cells.**
**a** Louvain clustering and projection onto wild-type 15 min UMAP, using variable genes as an input to define subpopulations. Cells are colored according to the subpopulation and the number is shown on top. **b** Heatmap shows expression of subpopulation-specific genes from the upregulated marker genes for each subpopulation defined in (**a**). Yellow colors indicate high expression and purple colors indicate low expression. **c** UMAP projection shows subpopulation representative genes with magenta for high expression and gray for low expression. **d** Distribution of the osmoconsensus signature expression induced for the indicated clusters. Solid black lines indicate the mean expression of the cluster while dashed black line indicates the population mean. Two-sided Wilcoxon test Benjamini–Hochberg adjustment of each cluster against the population is shown above each cluster. **e** Dot plot of 12 osmostress transcription factor activity across clusters. Dot size reflects the percentage of expressing cells; color indicates expression level (red = high, blue = low). **f** Scatter plot represents the single-cell co-expression of *STL1*, *ALD3*,

*CTT1*, *HSP12*, *GRE1*, *HXT5* (y-axis) and the induced osmoconsensus signature (x-axis). Cells are colored according to the Seurat cluster and dot size represents the expression levels of *HXT5* as a representative gene of the hyper-responsive population. **g** Scatter plot represents the average expression correlation for the genes in the induced osmoconsensus signature for hyper-responsive cells (y-axis) or the rest of the population. **h** Scatter plot of the percentage of cells expressing the induced osmoconsensus program for the hyper-responsive cells (y-axis) or the rest of the population (x-axis). Black line represents the line of equality. **i** Bar plot represents the cell growth of NaCl-sorted top 2% p*HXT5*-UbiM-mCherry-*tHXT5*-expressing cells (red) against NaCl-randomly sorted constitutive GFP-expressing wild-type cells. Cells were mixed at 1:1 ratio (time 0) and grown in the presence of 0.8 M NaCl for 48 h. The abundance of each population was determined by FACS. Stacked bar plots the mean and error bars the standard deviation (*n* = 3). Two sided t.test against the top population at t0 is shown. Symbols ns/*/**/***/**** represent *p*-values > 0.05, <0.05, <0.01, <0.001, <0.0001. Source data are provided as a Source Data file.

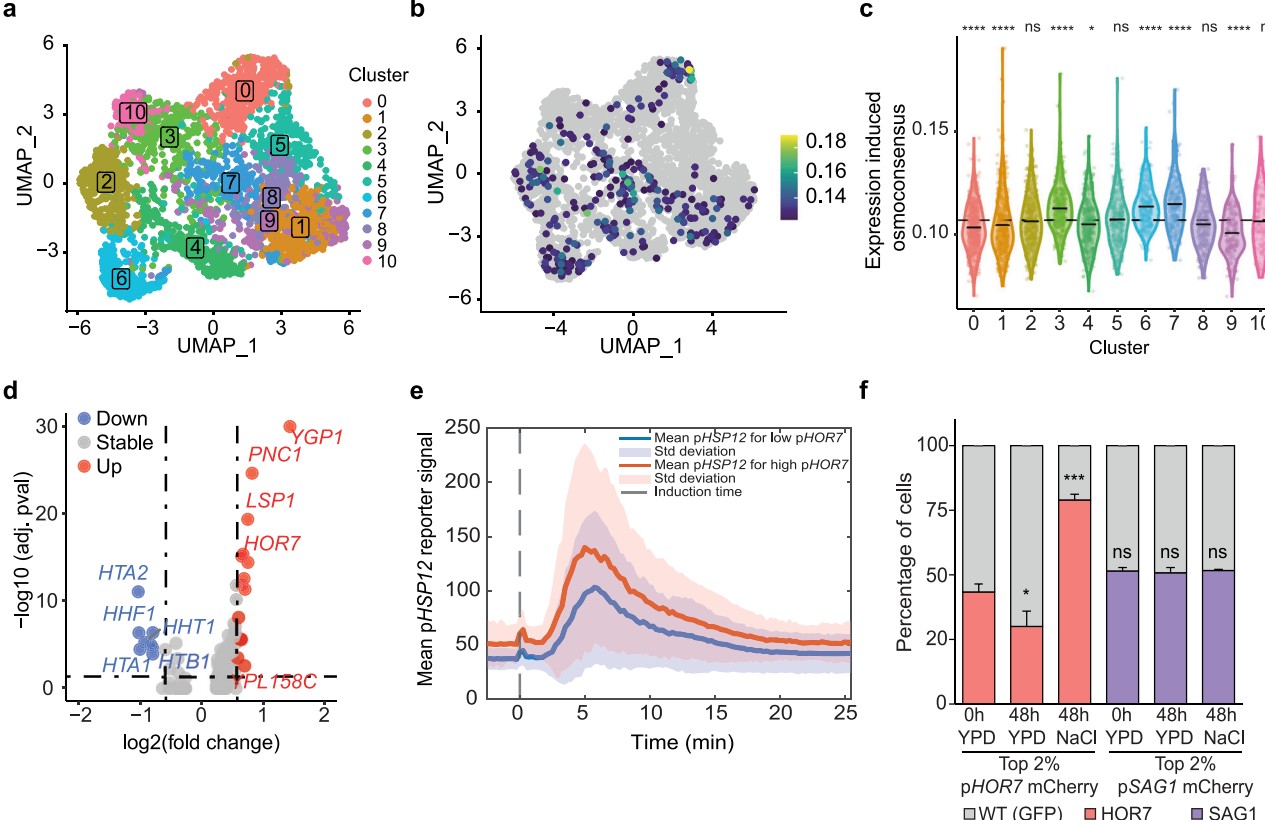

**Fig. 4 | Stochastic expression of stress programs in the absence of stress.**
**a** Louvain clustering and projection onto wild-type control UMAP, cluster number is shown **b** UMAP projection of cell with basal expression of the stress program. Highlighted cells are above the minimum cut of 0.11 and colored based on the expression level. Yellow indicates higher expression of the signature. **c** Distribution expression of the osmoconsensus signature induced for the indicated clusters in (**a**). Black lines represent the cluster mean expression and dashed line represents population mean. Statistical significance (Two sided Wilcoxon test, Benjamini–Hochberg adjustment) of each cluster against the population is shown above each cluster. **d** Volcano plot showing the number of genes differentially expressed (|log2(fold change)| ≥0.05 and *pval* ≤ 0.05) comparing basal stressed cells to the rest. Genes upregulated upon stress are shown in red and those downregulated in blue. **e** Dynamics of the p*HSP12* transcription site intensity labeled with PP7-mCherry in cells also bearing a p*HOR7*-MS2 reporter in the green channel. Cells were imaged for ten time points before

stimulation with 0.2 M NaCl at time 0 (dashed vertical line). Single cell traces are split in two sub-populations based on the level of the p*HOR7* transcription site intensity before the stimulus (red: High basal pHOR7 level *n* = 316, blue: low basal p*HOR7* level *n* = 526, n = 3). The solid line represents the mean of each subpopulation and the colored area the standard deviation between single cell traces. **f** Top 2% of pHOR7-mCherry and pSAG1-mCherry cells control conditions or after 1 h 0.4 M NaCl were isolated by Flow Cytometry and mixed at a 1:1 ratio with wild random sorted wild-type cells carrying constitutive GFP (10,000 cells/strain). This initial mixture (t0) was then grown in the in the indicated conditions. The fitness of each strain was determined by FACS after 48 h. Bar plot indicates the percentage of each strain. Error bars represent the standard deviation of three independent biological replicates. Statistical significance comparing the mean abundance to the reference timepoint (t0) is shown (two-sided t.test). Symbols ns/*/**/***/**** represent *p*-values > 0.05, <0.05, <0.01, <0.001, <0.0001. Source data are provided as a Source Data file.

correlation between transcriptional induction and repression. This revealed a global anti-correlation proportional to the stress response induction, observed only under stress conditions (Fig. 5a). This correlation was much weaker in *hog1* mutant cells, a finding that is consistent with their impaired response

(Supplementary Fig. 5a). Indeed, the anti-correlation was even more pronounced with the expression of ribosomal genes, which are known to be strongly repressed by osmostress, but not with the unresponsive genes (Supplementary Fig. 5b, c). The pattern was not affected by the expression level, as it was not detectable

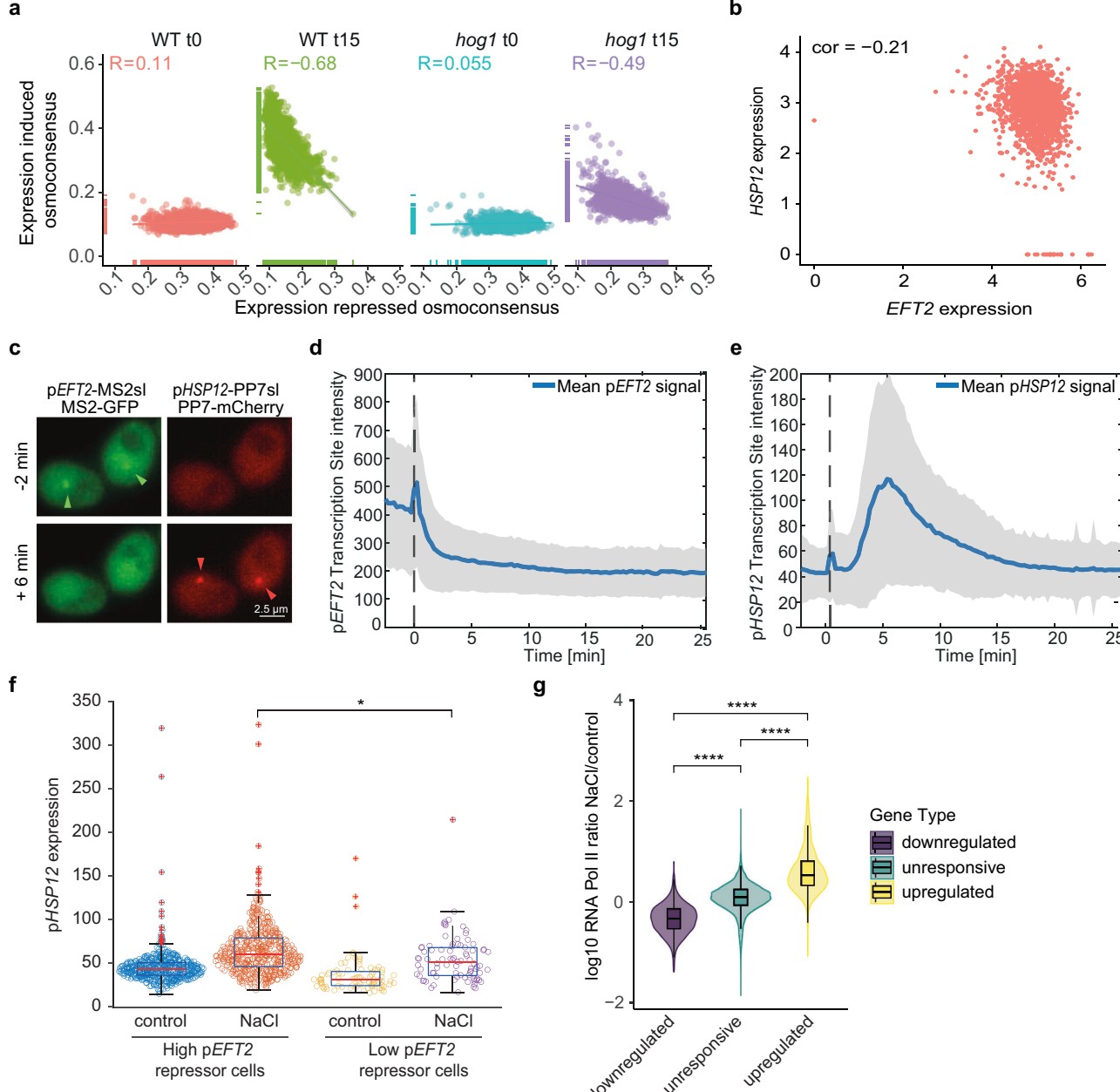

**Fig. 5 | Global transcriptome repression favors the induction of the osmoresponsive program. a** Per cell correlation between induced osmoconsensus score and repressed genes Points are colored by condition. Density plots show data distribution. Linear regression with 0.95 confidence interval and Pearson correlation are displayed. **b** Per cell expression correlation of representative pair for the induced osmoconsensus (*HSP12*) and repressed genes (*EFT2*). Pearson correlation value is shown. **c** Representative microscopy images displaying the transcription foci of the *EFT2* and the *HSP12* promoters labeled with MS2-GFPenvy and PP7-mCherry, respectively. In control, the green p*EFT2* transcription sites are visible (green arrowhead), upon stress, the red p*HSP12* transcription sites (red arrowhead) are detected (n = 4). Dynamics of the p*EFT2* (**d**) and p*HSP12* (**e**) transcription site intensities following a 0.2 M NaCl at time 0 (dashed line). Basal activity of p*EFT2* was detected and induction of p*HSP12* following stress was observed (35% of the whole population) (573 cells, n = 4). The solid line represents the mean and the shaded area the standard deviation. **f** These single cell traces were sorted in two sub-populations based on the level p*EFT2*

repression following stress. High p*EFT2* repressors show a drop of transcription site intensity of more than 50%. (489 cells) and display a higher inducibility of p*HSP12* compared low repression of p*EFT2* (84 cells). Boxplot represent 25th percentile (Q1) to the 75th percentile (Q3), with the central line indicating the median. The whiskers extend to data points within 1.5 times the interquartile range (Q3-Q1) (IQR). Data points outside this range are considered outliers and plotted with red crosses. Two-sided t.test is shown (*p*-value = 0.0031). **g** Violin plots show the distribution of RNA Pol II occupancy by ChIP-seq in bulk as the ratio of the normalized reads between NaCl (10 min 0.4 M NaCl) and control conditions from[5]. Genes are classified according to their bulk expression[23], osmoinduced (yellow, n = 677), osmorepressed (purple, n = 639), and other genes (green, serving as control n = 5136 genes). Boxplot displays summary statistics, median, interquartile range (Q1–Q3) and whiskers represent data points within 1.5 the IQR. Two-sided Wilcoxon test is shown. Symbols ns/*/**/***/**** represent *p*-values > 0.05, <0.05, <0.01, <0.001, <0.0001. Source data are provided as a Source Data file.

when we compared unresponsive genes, which were highly expressed (Supplementary Fig. 5d). Our scRNA-seq data support the hypothesis that global transcription repression is required for maximal gene induction upon stress.

We further explored this by identifying representative genes for both the induced and downregulated signatures. We selected *HSP12* and *EFT2* gene pair to generate *EFT2*-MS2-GFP and *HSP12*-PP7-mCherry mRNA reporters and track live single-cell transcription dynamics

(Fig. 5b, c). As expected, *EFT2* expression was high under control conditions and decreased immediately in response to stress (Fig. 5c, d), while *HSP12* expression was low under control conditions and transiently increased upon stress (Fig. 5c, e). We categorized *EFT2*-expressing cells into high repressors (>50% signal reduction) and low repressors (≤50% signal reduction) based on *EFT2* expression under stress. Tracing *HSP12* expression in these subpopulations revealed that high repressor cells showed a stronger *HSP12* induction compared to low *EFT2* repressors (Fig. 5f). This finding points to a direct relationship between the extent of general transcription repression and the induction of the osmoresponsive genes.

We hypothesized that transcriptional resources (e.g., RNA Pol II) from repressed genes are shuttled towards osmoresponsive genes upon exposure to stress. To explore this notion, we used available ChIP-seq data from the same time point and conditions (control and 0.4 M NaCl 15 min)[5] and calculated the ratio of RNA Pol II before and after stress in upregulated, repressed and unresponsive genes. As expected, stress-induced genes showed increased RNA Pol II, while repressed genes exhibited greater RNA Pol II loss compared to unresponsive genes (Fig. 5g). This suggests an interplay between transcriptional induction and repression, with repression potentially facilitating stress-responsive gene induction by freeing up transcriptional resources.

## A genetic screen identified key elements of the osmoadaptive response

To define the molecular logic underlying the transcriptional phenotypes of the osmostress response, we performed scRNA-seq profiling of mutants with transcription-related processes. To this end, we profiled a total of 325 mutants from of the Yeast Knockout Collection (YKOC) belonging to the gene ontology terms: chromatin organization; chromatin binding; chromatin remodeling; transcription regulation; transcription DNA template; RNA catabolism; and transcription factors (Fig. 6a). Additionally, we included mutants from the HOG pathway and a set of randomly selected mutants as a subset of control mutants. The YKOC, contains non-essential deletions in which the ORFs are replaced by a deletion cassette composed of a constitutive promoter (p*TEF1*), a resistance gene (G418), and a heterologous terminator and two non-transcribed barcodes upstream and downstream of the transcription unit (Uptag and Downtag)[27] (Supplementary Fig. 6a). We engineered RNA-barcoded gene deletions by replacing the G418 ORF and heterologous terminator with a *URA3* auxotrophic marker. This strategy preserved the Uptag barcode and constitutive promoter while shortening the original YKOC terminator (from 263 to 43 nt) and incorporating the Downtag genotype barcode into the *URA3* transcript's 3′UTR (Supplementary Fig. 6a). The resulting mutants allowed us to trace the genotype identity of complex cell mixtures by 3′-based polyA based scRNA-seq (see Methods). We individually transformed the strains from the YKOC and selected positive mutants by successive rounds of growth in selective media.

To determine the gene expression profile of each mutant, we grew each genotype individually in 96-well plates, with two wild-type strains as a reference. Cells were pooled before subjecting them to osmostress (control and 0.4 M NaCl 15 min) and methanol-fixed cells were used to generate droplet-based scRNA-seq libraries (10X Genomics) (see Methods, Fig. 6b). We profiled a total of 45,000 cells (25,610 control cells and 27,287 NaCl-treated cells) and performed a quality control filter to remove cell with an excess of molecules or with more than a single genotype barcode (see Methods). The resulting dataset contained 22,689 high-quality singlets and covered 260 distinct mutants with at least 6 cells in both condition. Each genotype was equally represented by an average of 34 and 43 cells in control and stress conditions (Supplementary Fig. 6b), respectively. We confirmed the robustness of our data using several metrics. First, we scored the expression of the osmoconsensus signature across genotypes

(Supplementary Fig. 6c) (Supplementary Data 7). As expected *hog1* and *pbs2* kinase mutants and *msn2* transcription factor mutants, were the mutants with the lowest expression (Supplementary Fig. 6c), consistent with bulk studies[2,5]. Finally, cells clustered in the UMAP space by condition (Fig. 6c) and not by genotype nor gene function (Fig. 6d, e). These results are also in agreement with the condition-dependent clustering observed in a 12-transcription factor deletion scRNA-seq study[28].

## Effect of genetic perturbations on the heterogeneity of the osmoadaptive program

In the scRNA analysis of the WT cells, we found that 6% displayed an increased expression of stress-responsive genes under control conditions (Fig. 4b and Supplementary Fig. 4c). To identify mutants with increased basal expression of stress-responsive genes, we established a threshold based on the top 10% of WT cells exhibiting higher osmoresponsive gene expression under control conditions. For each mutant, we calculated the percentage of cells exceeding this threshold, labeling those with at least 25% of cells as hyper-responsive. This approach allowed us to identify 33 mutants with a high frequency of basal stressed cells (Fig. 6f and Supplementary Data 7). To understand the nature of these regulators, we performed a physical protein-protein interaction network using only experimental evidence (Supplementary Fig. 6d). We detected multiple activities related to histone acetylation (Rpd3L, SAGA and Nua4 complexes), ATP-dependent chromatin remodeling complex SWI/SNF, and mutants related to the RNA Pol II and the Mediator complex (Supplementary Fig. 6d). Several subunits of the histone deacetylase complex Rpd3 Large (Rpd3L) were present within the top mutants that accumulated a large fraction of cells with an induced osmoconsensus program (30–63% of basal stressed cells in *rxt3* and *pho23* mutants, respectively) (Supplementary Fig. 6d). Indeed, mutants of histone deacetylase Sin3 and Nua4 complexes (*yng2*), the SWI/SNF chromatin remodeling complex (*snf5*), and the transcriptional regulator (*spt21*) displayed higher expression of the p*HOR7*-mCherry-t*HOR7* reporter under basal conditions (Fig. 6g, Supplementary Fig. 6e). Hence, our data suggest that impaired homeostasis of histone acetylation by Rpd3L or unstructured chromatin structure caused by the mutation SWI/SNF allows basal input-independent gene expression. Of note, some of these activities have been reported to be relevant for the induction of osmostress program[4,29], suggesting a dual role of these proteins in the switch between repressed to induced states.

We next applied the same rationale to study the genetic logic hyper-responsive cells. To this end, we calculated the percentage of cells in each mutant with higher expression of stress-responsive genes in response to stress. We identified 44 mutants that had at least 25% of hyper-responsive cells (Fig. 6h) (Supplementary Data 7). The physical protein-protein interaction network showed that regulators span multiple functions (Supplementary Fig. 6f), including expected ones such as chromatin remodeling (i.e. *ies4*, from the INO80 complex)[30,31], histone deacetylases (e.g. *hsa1*) and *bem1*, a scaffold protein upstream from the Sho1 branch of the HOG pathway previously reported to display stronger response than the wild type[12,32]. Additionally, within the network, several unexpected functions were observed, transcription silencing proteins (*sir1*, *sir3*), RNA Pol II regulators (*ctk2*), HOG signaling, and other factors linked to the induction of the osmostress program, such as transcription factors (*rtg1/rtg3, cin5*)[33–35] and chromatin remodeling complex RSC (*rsc1, npl6*[4,11]) (Supplementary Fig. 6f). In yeast, the RSC complex, recruited by Hog1 to osmoresponsive genes, facilitates induction by removing nucleosomes. Conditional RSC alleles (Rsc9[ts]) and SAGA complex mutants (*gcn5*) exhibit bimodal stress-response expression at 0.4 M NaCl, where WT cells remain unimodal. However, mutants like *rsc1*, *gcn5*, and *hda1* display subpopulations with higher expression under the same conditions by scRNA-seq[12]. To validate the hyper-responsive subpopulations, we

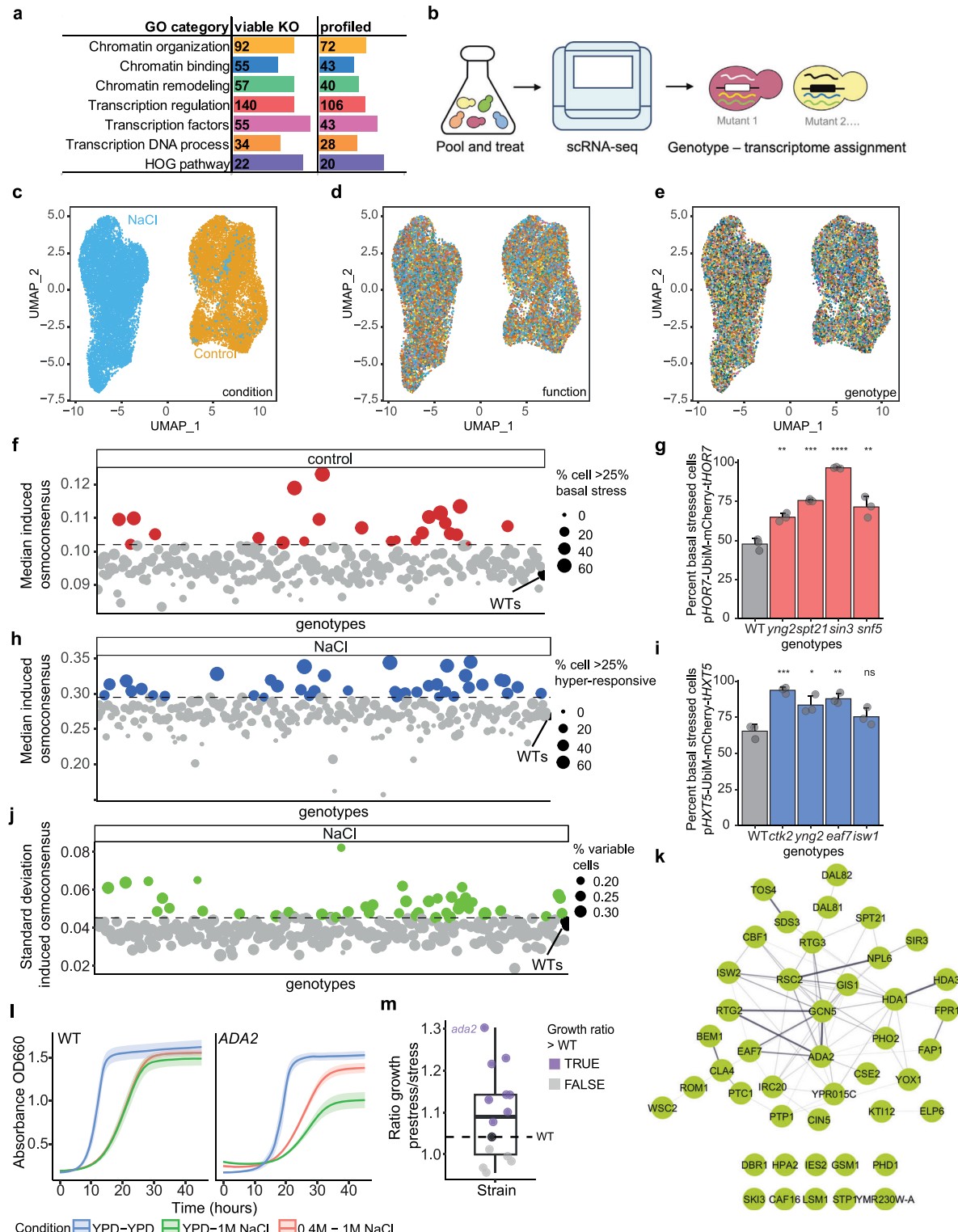

integrated the p*HXT5*-mCherry-t*HXT5* reporter in 4 of these regulators, which included ATP-dependent remodeling complexes (*isw1*), histone modifiers (*eaf7* and *yng2*), and an RNA Pol II regulator (*ctk2*). We measured p*HXT5* by FACS and detected that all mutants showed a higher frequency of *HXT5*-expressing cells, thus validating the scRNA-seq data (Fig. 6i, Supplementary Fig. 6g). Hence, our data point to a multifunctional network that limits the appearance of hyper-responsive cells within a population.

Our analysis identified a subpopulation of mutants with a hyper-responsive phenotype including expected (SAGA and RSC) among

other unknown regulators. Indeed, we observed that mutants with hyper-responsive populations exhibited increased variability in the osmoconsensus program (Supplementary Fig. 6h, 6i). To identify regulators of osmoresponsive heterogeneity, we scored the standard deviation of each mutant in the stress condition. We found 44 mutants (16% of mutants) whose deletion led to an increase in the heterogeneity of the induced osmoconsensus program (Fig. 6j, k, and Supplementary Fig. 6j), which correlates with the Fano factor (Supplementary Fig. 6k) (Supplementary Data 7). At the level of gene function, the signature exhibits comparable expression variability

**Fig. 6 | A genetic screen served to identify key elements of the osmoadaptive transcriptional phenotypes. a** Distribution of the selected mutants for targeted Perturb-seq. Columns indicate the total number of genes in the GO term and the manually selected HOG pathway. The number of viable and profiled mutants from the YKOC are shown including mutants belonging to multiple categories. **b** Schematic representation of the experimental design. Briefly, the selected mutants were grown individually and pooled and harvested in the absence of control and stress (15 min NaCl 0.4 M). UMAP of the entire dataset colored by treatment (**c**), GO class (**d**) genotype (**e**). **f** Basal expression distribution of the induced osmoconsensus signature across mutants with ≥6 cells. Point size reflects the percentage of cells per mutant exhibiting high expression (≥90% of wild-type). Mutants with ≥25% of high expressing cells over the wild-type cells are highlighted in red (dotted line). **g** Barplot represents the mean and standard deviation expression of basal stress- marker (pHOR7-UbiM-mCherry-tHOR7) by FACS for the selected mutants from (**f**) (n = 3). Two sided t.test against the wild type (gray) is shown. **h** Expression of osmoconsensus signature across genotypes under stress. Point size indicates percentage of hyper-responsive cells; highlighted points show mutants with ≥25% of hyper responsive cells (wild type in black). **i** Expression of the hyper-responsive marker (pHXT5) by FACS for hyper-

responsive mutants selected from (**h**). Bar plot represents the mean expression and standard deviation (n = 3). Basal expression is determined using the wild type strain as a reference (gray bar). Two side t.test is shown against the wild type. **j** the variance of the induced osmoconsensus signature for each strain; color points and dotted line selected strains. **k** Physical protein-protein interaction network of mutants showing high heterogeneity (n = 44, from (**j**)).The network was generated using STRING and based on experimental evidence, and connection between nodes represent confidence values above 0.150. **l** Growth dynamics of wild type and *ada2* strains in control conditions (blue) or in 1 M (green) or pre-stressed and 0.4 M NaCl for 30 min and then grown in 1 M (red). Solid line represents the mean and ribbon represents the standard deviation (n = 6–7). **m** Boxplot represents the distribution of growth ratios (prestress/stress) across different strains, showing the median, interquartile range (Q1–Q3), and individual data points colored to indicate ratios respect to the WT threshold (black dot and dashed line). Mutants shown in purple exhibit a higher growth ratio (fitness) than the wild type when prestressed (*ada2, bem1, cla4, fap1, hda3, ies2, ptc1, ski3*), than mutants in gray (*lsm1, dbr1, npl6, rtg3, pdh1*) compared to the WT (n = 6–7). Symbols ns/*/**/***/**** represent *p*-values > 0.05, <0.05, <0.01, <0.001, <0.0001. Source data are provided as a Source Data file.

across its associated GO categories (Supplementary Fig. 6l). To gain a global understanding of the nature of the transcriptional mutants that impact heterogeneity, we generated a physical protein-protein interaction network (Fig. 6k). Interestingly, *gcn5* emerged as a central node, connected to *ada2*, both components of the SAGA histone acetyltransferase complex and also subunits of the RSC complex. Of note, SAGA and RSC have been previously identified as drivers of bimodal gene expression upon osmostress using single gene reporters[12]. These findings reinforce the role of chromatin remodeling as a driver of heterogeneity in the expression of the entire signature and the potential of our dataset in identifying uncharacterized regulators. Additionally, we found several unknown mutants, including RNA binding proteins (*lsm1, ski3*), regulators of histone post-translational modifications (*yox1, hpa2*, sir3, *eaf7, hda1, hda3*), transcriptional activators or repressors (*spt21, gis1, yox1, dal81, dal82*), and several transcription factors (*rtg3, rtg3, pho2, phd1*), that showed an increase in transcriptional heterogeneity when mutated. Overall, our data suggest that transcriptional heterogeneity in response to stress is modulated through several mechanisms, which range from direct transcriptional regulators to signaling proteins.

We assessed the phenotypic consequences of transcriptional heterogeneity by focusing on Ada2. Despite *ada2* mutant shows a low overall expression of the osmoconsensus (Supplementary Fig. 6c), it exhibited significant heterogeneity, with some cells expressing the osmoconsensus higher than wild type (Supplementary Fig. 6j). We next tested if pre-stressing variable mutants such as *ada2* with a mild osmostress pulse (0.4 M NaCl 30 min) could provide a beneficial effect to higher osmostress[36–38]. To do so, we compared the growth of wild-type and *ada2* mutant strain under control or high osmolarity (1 M NaCl), with and without pre-exposure to osmostress (0.4 M NaCl 30 min). As anticipated, wild-type cells exhibited a significant growth delay when switched from control conditions to 1 M NaCl and displayed a minimal effect on growth if pre-stressed (Fig. 6l). Remarkably, *ada2* cells, which displayed a slower growth when directly exposed to 1 M NaCl, showed an enhanced growth when compared to the wild type when pre-stressed (Fig. 6l). Similarly, pre-stressing *ada2* cells led to a 2.2-fold increase in pHXT5-Ubi-mCherry-tHXT5 expression, while wild-type cells exhibited only a 1.3-fold increase when compared to non pre-stressed cells (Supplementary Fig. 6m). We then investigated whether variable mutants exhibited enhanced fitness when pre-stressed. We monitored the growth of the 13 most variable mutants (Fig. 6j), encompassing diverse functions including chromatin regulation (see Methods). We calculated the ratio of endpoint growth between pre-stressed and cells directly switched to 1 M osmostress. Notably, 8 out of 13 mutants (61.5%) demonstrated faster adaptation

compared to the wild type (Fig. 6m). This observation suggests a pattern where variable mutants display improved fitness when prestressed, indicating a change on adaptive fitness. These findings suggest that stress-induced transcriptional heterogeneity can generate sufficient cell variability within the population to prime the emergence of long-term stress-resistant phenotypes.

## Discussion

Our study combines longitudinal scRNA-seq with a transcription-focused single-cell perturbation screen to build a high-resolution transcription map of the adaptive gene expression landscape. The canonical model for stress-activated expression assumes "bulk" behavior where sets of genes are upregulated or downregulated upon stress. Single-cell reporter measurements suggested that homogenous activation of Hog1 SAPK upon stress leads to a heterogeneous transcriptional output[12]. Here we showed that, even at the peak of the expression, the structure of the osmoadaptive program shows high heterogeneity: with only a core set of genes (25%) universally used and a mean response per cell that involved only half of the responsive genes revealing that cells explore multiple adaptive strategies. A small subset of hyper-responsive cells used more genes during stress, showing enhanced fitness. Notably, these cells exhibited reduced fitness under normal conditions, indicating that the number of stress-responsive genes must be carefully controlled to prevent transcriptional burden and maintain cellular fitness. Induction of stress-responsive program seems to be limited by transcriptional resource availability. Osmostress causes transient dissociation of chromatin-bound proteins, leading to genome-wide transcriptional inactivation[5]. This releases transcriptional resources simultaneously with Hog1 activation and target gene association, potentially favoring transcriptome rewiring. Our data indicate RNA Pol II machinery as a critical limiting factor in the response, revealing a proportionality between induction and repression. Stronger global genome repression correlates with increased transcription, more responsive genes per cell, and higher RNA Pol II occupancy. Our longitudinal study demonstrates that repression is crucial for induction stress response, suggesting that cells engage in one transcriptional program at a time, as no cells trigger adaptive responses without accompanying repression.

Stress-responsive gene usage reveals a modular cellular response. About 30% of wild-type cells show a heat stress-like signature, typically masked in bulk measurements[23]. An additional module with metabolic and oxidative stress genes suggests diverse transcriptional stress adaptation paths. Stress cross-protection in yeast involves improved tolerance from initial adaptive responses[37,39–42]. Msn2 overexpression or expression of ESR genes (*CTT1* and *TSL1*) provides multi-stress

resistance, though an undefined general mechanism[18,43–45]. Our data imply that the modular expression of stress genes could create specialized cell subpopulations that enable population stress cross-protection but also suggest that not all cells will acquire tolerance to the same stresses. We also found that heterogeneity extends beyond individual genes and involves different transcription factors, generating subpopulations with differential adaptive potential. This mechanism may explain persister-like phenotypes in clonal populations.

Under basal conditions, histone acetylation levels are actively controlled, but excessive acetylation may derepress the expression of the osmoresponsive genes. Negative regulators of the stress responses have often been overlooked. Our findings expose an unexpected multifunctional negative regulator network involving osmoadaptation components. Notably, mutants promoting hyper-responsive populations demonstrate significant heterogeneity, underscoring the role of negative regulators to shaping the adaptive response.

Bimodal gene expression is a globally conserved example of transcriptional heterogeneity. In *S. cerevisiae*, bimodality has been measured through gene-specific reporters and is often described as a property of stress-responsive genes (e.g., diauxic shift and osmostress)[12–15]. Partial nucleosome eviction in osmoresponsive genes through the impaired function of Gcn5 and Rsc9 (SAGA and RSC complexes, respectively) is one of the determinants of this bimodal behavior[12]. In our targeted genetic perturbation profiling, *gcn5* scored as one of the most heterogeneous mutants, along with its interactor *ada2*, a coactivator of the SAGA/ADA complex, which promotes Gcn5 acetyltransferase activity[46]. Additionally, we identified subunits of the Nua4 histone acetyltransferase complex (*EAF7*), as well as the catalytic subunit of histone deacetylase complex (*HDA1*) and the subunit required for its activity (*HDA3*)[47,48]. Additional regulators included a variety of functions regulating the heterogeneity of the osmoresponsive program, ranging from transcription factors, RNA decay, mitochondrial retrograde signaling, to upstream and downstream signaling proteins[5127]. We examined the phenotypic consequences of transcriptional heterogeneity in variable mutants like *ada2*. These mutants showed reduced growth under normal conditions but increased phenotypic plasticity, enabling greater adaptability to extreme stress. This enhanced flexibility may explain the emergence of persistent cells in response to treatments in highly heterogenic cell populations. Therefore, our study provides a time-resolved single cell-resolved map of the transcriptional landscapes underlying adaptive phenotypes and reveals genetic mechanisms that regulate them.

## Methods

### scRNA-seq strains
For the longitudinal profiling of wild type (BY4741) was used as parental strain to which *hog1*::NAT deletion was performed using PCR tagging. Both strains were used to generate C-terminal tagged *TDH3*-9myc::KAN and *TDH3*-9myc::HPH respectively. Standard yeast transformation was done using the LiAc method into the corresponding yeast background and colonies were selected by marker selection and colony PCR. PCR cassettes were obtained using primers in Supplementary Table 1 using the yeast PCR toolbox plasmids as a template[49].

For the generation of targeted Perturb-seq strains, frozen glycerol stocks from the haploid yeast knock out collection (YKOC) were grown on YPD (Yeast Peptone Dextrose medium) supplemented with G418 (Geneticin, 200 mg/L). Strains were individually transformed and selected in *URA3* plates. Two BY4741 strains were manually transformed to integrate the barcoded pTEF1 *URA3* construct containing the Downtag barcode, these strains were validated by sequencing. The primers used to modify the YKOC and the corresponding wild type strains are listed in Supplementary Table 2.

### Fluorescent reporters
Recombinant DNA techniques and transformation of bacterial and yeast cells were performed using standard methods. To generate reporters for cell states, we used the MoClo Yeast Toolkit Modular cloning system[50]. Building of the plasmid constructs was achieved using Golden Gate assembly. Each reporter contains a transcription unit composed of the corresponding promoter (700 bp upstream of the annotated ATG), UbiM degradation signal, florescent protein, and terminator (300 bp downstream of annotated STOP codon). All part sequences were either mutated or synthesized to avoid of the BsmBI, BsaI, and NotI recognition sequences. Promoter and terminator sequences were amplified from BY4741 genomic DNA and purified using the MiniElute PCR purification (28004, Quiagen). Plasmids generated in this study are described in Table 3.

### Live mRNA tracking strains
The strains and plasmids used for nascent transcript monitoring are included in Supplementary Tables S2 and S3. All strains are derived from the *Saccharomyces cerevisiae* W303 background (Ralser et al., 2012). The nuclear marker was created by tagging Hta2 with the tdiRFP protein with a TRP or a NAT marker in MATa or MATα cells, respectively (Wosika et al., 2016). The strains were then transformed with the PP7-mCherry (MATa) or the MS2-GFPenvy (MATα), expressed under the constitutive *ADH1* promoter (Wosika & Pelet, 2020). In these strains, PP7 or MS2 stem loops, regulated by a promoter of interest, were integrated in the GLT1 locus (Larson et al., 2011; Wosika & Pelet, 2020). The promoters (−1000 to 0 before ATG) were amplified form W303 genomic DNA and cloned in front of the stem loops. The length of the stem loops integrated in the genome were verified by colony PCR. The MATa and MATα cells were mated on YPD plates and the diploid strains were isolated on selective plates (SD-HUT+NAT).

### Live mRNA tacking analysis
Cells were inoculated in SD-full medium (Complete CSM DCS0031; ForMedium) and grown overnight until saturation. The culture was then diluted in fresh medium and maintained in log-phase growth (OD < 0.4) for 24 h through successive dilutions before imaging. 200 μl of cell suspension at OD 0.05 was loaded in the well of a 96-well plate (PS96B-G175; SwissCI) that had been pre-coated with Concanavalin A (L7647; Sigma-Aldrich). Imaging was carried out on a Nikon Ti2 inverted microscope housed in a temperature-controlled incubation chamber set to 30 °C, with micro-manager software controlling the system (Edelstein et al., 2010, Ch. 14, Unit14.20). Fluorescent excitation was provided by a Lumencor Spectra III light source. For transcription site measurements, the LED intensity was reduced respectively to 20% (GFP) and 25% (mCherry) of the maximum power to minimize photo-bleaching. Cells were imaged using a 40X oil objective, a quadruple band dichroic (DAPI/FITC/Cy3/Cy5, F68-400; Chroma), and appropriate emission filters. Images were captured with a Hamamatsu ORCA-Fusion sCMOS camera.

Using a piezo stage (Nanodrive; Mad City Lab City), five Z-planes were recorded (−1 to +1 μm) in the fluorescent channels (FITC and Cy3), with transcription sites recorded every 15 seconds. The nuclear marker (Cy5) and brightfield images, used to segment the cells, were captured at every third time point. At each time point, up to five XY positions in one well were imaged, with focal plane accuracy ensured by hardware autofocus. Before the tenth time point, the acquisition was paused to add 100 μl of stimulation medium concentrated threefold to reach a final concentration of 0.2 M NaCl.

### Segmentation and data analysis
The recorded time-lapse measurements were analyzed using the YeastQuant platform[51]. Brightfield segmentation was performed with CellPose based on the cyto2 model[52,53]. The detected cell objects were then combined with an intensity-based segmentation of the nuclear

marker to define a nucleus and cytoplasm objects for each cell. An ExpandedNucleus object is defined by dilating the nucleus by 5 pixels.

Two additional objects are defined to quantify the transcriptional activity of the different promoters driving the stem loop transcripts. The HighPix object represents the 10 brightest pixels in the expanded nucleus of each individual cells in the fluorescent channel of the phage coat protein. The ConnectedHighPix object verifies the local connectivity of these high intensity pixels. If at least 5 pixels are connected, the ConnectedHighPix object is defined. Three consecutive ConnectedHighPix must be detected to determine that a cell is actively transcribing. If three consecutive ConnectedHighPix are observed before the induction, cells are considered as basally transcribing.

The transcription site intensity is calculated as the difference between the mean intensities of the HighPix object and of the expanded nucleus. The identification of the ConnectedHighPix allows to determine when transcription starts and ends. During this window of time, the peak intensity can be measured as the difference between the maximum of the transcription site intensity trace and the basal level. The basal level corresponds to the mean intensity of the 10 points before the stimulus.

The basal level of the p*HOR7* reporter was used to split the cells in two sub-populations. Cells where the basal level was above the mean basal level +1x the standard deviation of the whole population were categorized as high basal cells.

To characterize the repression of the p*EFT2*-MS2 cells, basally transcribing traces were selected. The transcription site intensity measurements was normalized (normalizedData = (celltrace − minval)/(basalactivity − minval)). If the drop between the basal and post-induction levels in the normalized trace is >50%, the cell belongs to high repression sub-population.

## Competition assays

Wild type strains carrying the indicated expression reporters were grown to mid exponential log phase ($OD_{660}$ = 0.6) in Synthetic Complete Media (SCM) and sorted using Aria SORP in the absence (control) or presence of stress (0.4 M NaCl 1 h) (Becton Dikinson). A total of 20,000 cells of each top 2% of the population was sorted and 20,000 cells of a random sorted wild type strain carrying a constitutive GFP (p*TEF1*-YumuGK1) and grown in the same conditions was sorted on top in a final volume of 200 µl of rich media (YPD). A total of 150 µl of the mixed culture was fixed with sodium azide (S2002, Sigma-Aldrich) as a fixating agent that does not disturb fluorescence of the reporters as time 0. The remaining culture was evenly split and incubated in YPD and YPD 1 M NaCl and grown at 30 °C. Cells were diluted every 12 h and fixed after 48 h from t0. Abundance of each population was determined by flow cytometry (see below).

## Flow cytometry analysis

Cells were recorded from each sample according to their FSC and SSC distributions and unmixed to identify the fluorescence signal for each fluorophore (mCherry or GFP). For competition assays, cells were gated based on the constitutive expression of GFP of the wild type strain versus the side scatter for three biological replicates.

To read the expression of destabilized fluorescent reporters (p*HOR7*-UbiM-mCherry-t*HOR7* and p*HXT5*-UbiM-mCherry-t*HXT5*), the full spectrum of 1000 cells were recorded Cytek® Aurora (4-laser and 64 Fluorescence Emission Detection Channels) gated according to the FCS and SSC distributions. Cells were grown in selective media (SCM) to mid exponential phase and subjected or not to osmotic stress (0.4 M NaCl). Fluorescence was measured before and after stress (0.4 M NaCl 1 h). The unmixed signal was used to assess the expression distribution of each mutant against the wild type. To calculate the basal signal of the unstained strain was used as a reference and the expression of the wild type strain carrying the p*HOR7*-UbiM-mCherry-t*HOR7* reporter. For mutants carrying the same reporter, the basal

fluorescence was calculated using the wild type strain carrying the reporter as a reference. For hyper responsive cells, the expression of wild type cells carrying the p*HXT5*-UbiM-mCherry-t*HXT5* was determined by comparing the expression of the strain upon stress (0.4 M NaCl 1 hour) to its expression in control conditions. For mutants carrying the same reporter, we assessed the expression upon stress (0.4 M NaCl 1 h). Cytometry data were analyzed using FlowJo™ Software (BD Life Sciences). Per each strain we used three independent biological replicates which are represented in the standard deviation bars. Statistical significance is shown using a paired t.test.

## Growth curves and reporter expression in pre-stressed conditions

Indicated cells were grown overnight rich media. The next day cells were diluted and allowed to recover to mid exponential phase. Then cells were diluted at an $OD_{660}$ of 0.025 and pre-treated or not to 0.4 M NaCl for 30 min (pre-stressed condition). Then control and pre-stressed cells were grown in YPD or YPD 1 M NaCl at 30 °C for 50 h. Growth was monitored using a Synergy HXT instrument by reading absorbance at 660 nm every 30 minutes. In each biological replicate (n = 7, except for *CLA4*, n = 6). For each genotype we calculated the mean absorbance of the prestressed and 1M NaCl using all the biological replicates for each condition. To calculate the ratio we divided the prestressed or not prestressed. The ratio of the wild type was used as a reference to identify mutants with higher or lower ratio. To measure the effect of pre-stressing on gene expression we integrated the p*HXT5*-UbiM-mCherry-tHXT5 reporter in wild type and *ada2* mutant. Cells were grown to mid exponential log phase in SCM and pre-stressed or not with 0.4 M NaCl. Then cells were shifted to SCM 1M NaCl for 1.5 h before the addition of Cycloheximide (0.1 mg/ml) (C4859-1ML) and incubated for 1 h. The fluorescence of 5000 per sample was measured by flow cytometry Cytek® Aurora and analyzed as described above. For each strain and condition (pre-stressed or not) the median expression of the population was extracted using the FlowJo software. The difference in expression between conditions was calculated by ratio of *HXT5* signal of pre-stressed/ not pre-stressed in which values higher than 1 indicate higher expression of pre-stressed cells. The mean ratio of 3 independent biological replicates was used to calculate the difference between strains (Wilcoxon test).

## Cell growth and harvesting

For the wild type and *hog1* dataset, each strain was grown individually overnight in YPD. The day of the experiment, cells were diluted to $OD_{660}$ 0.05 and cells were allowed to grow to mid exponential log phase $OD_{660}$ 0.6. Cells were pooled at a 1:1 ratio in a before being or not subjected to osmostress 0.4 M NaCl for 0, 5, 15, and 30 min. At each time point cells were pelleted by centrifugation 1 min at 3000 rpms, and pelleted cells without media were immediately resuspended with ice cold 80% (Scharlab, ME0301005P).

## Library preparation

For both datasets, methanol fixed cells were rehydrated as suggested by 10X genomics instructions. Briefly cells were allowed to equilibrate on ice for 20 min and then pelleted by centrifugation 3 min at 3000 rpm. Wells were washed twice with DPBS 1X (Thermo Fisher, 14190144) with 0.04% BSA (Thermo Fisher, AM2616) and diluted in the same media at a final concentration of 1000 cells/µl that was used for chip loading. Each time point (0,5,15,30) was loaded into an individual lane. In the deletion-based scRNA-seq profiling, we loaded control samples into one lane, and NaCl-treated samples into two lanes since NaCl-treated samples typically have a lower recovery rate. Samples were processed following manufacturer protocol CG000315 Rev C (10X genomics) with a minor modification. Due to the presence the cell wall 11 µl of Zymolyase T (100 mg/ml) was added to the cDNA reaction mix and loaded into the chip.

A total of 9 PCR cycles were performed for each sample yielding a concentration of 1–2 ng/µl after bead purification. Amplified cDNA was quality controlled on the Bioanalyzer 2100 system. Single Cell 3' Gene Expression libraries were generated using 10 µl (25%) of the total cDNA (40 ul) following manufacturer's protocol. A total of 13 PCR cycles were performed to generate dual index barcoded libraries. After purification library concentration was measured by Qubit and expected library size distribution was confirmed using the Bioanalyzer 2100 system. For sequencing libraries, all samples from each dataset—whether longitudinal or deletion-based scRNA-seq—were combined and sequenced. The longitudinal assay was sequenced using a NovaSeq S4 while the deletion-based scRNA-seq was sequenced using a NextSeq500.

### Read pre-processing and alignment

Resulting FATSQ files were processed using Cell Ranger (v4.0.0)[54] using default parameters. Reads were aligned using the reference genome from[55]. Additionally for the wild type and *hog1* mutant dataset the sequence of the resistance markers *KAN*, *NAT* and *HPH* were added as separate chromosome to the reference genome. For the Perturb-seq experiments the Downtag barcodes were downloaded from (http://www-deletion.stanford.edu/YDPM/YDPM_index.html) and added as a separate chromosome to the reference genome with the corresponding barcode (bc-systematic name). The Yeast Knock out collection contains replacement strains. To avoid naming conflicts with the aligner, we named repeated genotypes in the YKOC with sub-indexes (-1, -2 or -3 for the successive repeated genotypes in the YKOC). Wild type strains YMN478 and YMN479 barcodes (Supplementary Table 2) were added manually to the barcode list. With the following structure:

> bc-Systematic Name
> NNNNNAACGCCGCCATCCAGTGTCGAAAACGAGCTCGAATTCAT
> CGATNNNNNNNNNNNNNNNNNNNNNNNCTACGAGACCGACACCG

### Data processing

**Genotype assignment.** For the longitudinal scRNA-seq dataset we used a genotype specific marker gene-expression based assignment (Supplementary Fig. 1a) using the following criteria. Initial quality control was applied to remove low-quality cells and potential doublets. Cells with fewer than 500 or more than 3000 detected genes were excluded from further analysis. Then, genotype assignment was based on the expression of specific genetic markers using Loupe Cell Browser (10X Genomics). Wild-type (WT) cells were identified by the expression of *KAN* (>1 counts) and the absence of *NAT* and *HPH* expression (0 counts). For *hog1* mutant cells were identified by the expression of *HPH* and/or *NAT* (>1 count) and the absence of *HOG1* expression (0 counts). The assigned cell genotypes were then imported into the metadata of each Seurat[56] object. Cells not meeting either of these criteria were excluded from further analysis.

For the deletion based scRNA-seq of transcription mutants, we scored the expression of all possible genotype barcodes. We implemented a stringent assignment systems un which cells were only assigned and kept for analysis. Cells with no detectable expression of a genotype barcode were labeled as "Unassigned" and cells with more than one detected barcode were labeled as "Doublet", which were both removed from the analysis. Additionally, genotype barcode sequences were removed from the expression matrix before proceeding to downstream analysis.

To perform the downstream analysis, the Cell Ranger outputs of each dataset were used to generate the corresponding cell expression matrices by combining all samples of each dataset. We generated a total of 6 Seurat objects. The complete longitudinal experiment contains merged Seurat objects of the time course (times 0, 5, 15 and 30) including both wild types. From this experiment we performed individual analysis by generating individual objects of wild type time 0 and time 15 min, and *hog1* mutant cells time 0. Finally, for the deletion-based scRNA-seq we merged the time 0 and time 15 samples into a single dataset.

### Data normalization and clustering

To normalize gene expression, for all datasets (longitudinal and deletion scRNA-seq) we removed ribosomal protein genes (RPL and RPS) and used the standard Seurat processing guidelines. We normalized data using the NormalizeData function from Seurat (Seurat v4.0)[56]. We used previously reported cell cycle variable genes in scRNA-seq studies[28] to score the cell cycle stage using the CellCycleScoring function. Calculated cell cycle scores were used to regress out the cell cycle effects using the vars.to.regress function and the calculated cell cycle scores. We also identified highly variable genes using the FindVariableFeatures nfeatures = 200.

To perform cell clustering, first we performed a linear dimensional reduction using the "RunPCA" function from Seurat Package using PC1 and PC2. For the deletion-based scRNA-seq we then then applied the Seurat pipeline FindNeighbors (dims 1:10) and FindClusters (resolution = 0.5). To visualize the UMAPs we used RunUMAP (dims 1:10).

To extract cell state markers, we applied the differential expression function included in Seurat through FindAllMarkers for the complete dataset (both conditions) and each condition individually. Gene ontology enrichments of upregulated cell state markers were performed using Metascape v3.5.20230501[57] default parameters using *S. cerevisiae* as a specie (markers and Gene Ontology results). To score the expression of gene signatures (see below for information of gene lists) we used the UCell package AddModuleScore_UCell[58].

### Gene signatures

A list of genes used in each signature is available as Supplementary Data 1.

For induced, repressed and unresponsive genes, we used already available OsmoAtlas dataset[23]. This dataset represents a consensus expression comprises 5 independent RNA-seq experiments.

**Induced genes.** This signature represents the top 200 genes classified as upregulated (FC ≥ 2 and *pval* < 0.05).

**Repressed genes.** This signature represents the bottom 200 genes classified as downregulated (FC ≥ −2 and *pval* < 0.05).

**Unresponsive genes.** This signature represents a set of 200 genes classified as unresponsive (FC −0.5 to 0.5 and *pval* < 0.05).

**Representative signature genes.** To identify genes whose expression correlated with the induced osmoconsensus (Upregulated 200), repressed (Downregulated 200) or unresponsive (Unresponsive 200), we calculated the Pearson correlation of each gene in the expression matrix against both signatures.

To generate the transcription factor signatures, we extracted the positive regulated targets genes from SGD (www.yeastgenome.org/). For Msn2/4 that overlapped with the upregulated genes in the OsmoAtlas[23].

Daughter cells: This signature represents a set of genes curated from the literature and SGD whose expression is specific for daughter cells[59,60].

To calculate the mean, median, standard deviation and variance of each gene signature was performed in R using the: mean(), median(), sd(), var() functions respectively. Each signature whose expression was calculated using the UCell package and stored in the metadata (#ref). To calculate the Fano factor (defined as the variance-to-mean ratio). The gene signature expression data was extracted from the metadata and grouped by sample and condition for the longitudinal daset and for the genotype and condition for the targeted perturbation dataset.

the Fano factor was calculated using this formula: Fano Factor = Variance of Expression/Mean of Expression.

## Subpopulation definition

To identify subpopulations with specific transcriptional phenotypes, we used an expression-based threshold using induced osmoconensus signature. For basal stressed cells in the wild type time 0 (longitudinal dataset), the last decile of expression (0.9) from the expression of the Upregulated 200 signature. For the detection of hyperresponsive genes we calculated the co-expression of *STL1*, *ALD3*, *CTT1*, *HSP12*, *GRE1*, *HXT5* using the Plot_Density_Joint_Only function from the scCustomize package (v0.6.1)[61]. We extracted the co-expression values of each cell to add the values as "Upreg_coexp" column in the wild type time 15 min object metadata. Cells whose expression is higher than the mean plus the Gmd were considered as hyperresponsive.

For the deletion based scRNA-seq dataset, we defined basal stressed population using as a threshold the last decile of expression (0.9) from the expression of the Upregulated 200 signature of the wild type cells included in the same experiment. We calculated this threshold for wild type cells as control and NaCl conditions separately. Then for all detected mutants we calculated the percentage of cells above their respective threshold. For the analysis we only considered cells with at least 6 cells, but we reported the values for all detected genotypes (Supplementary Table 7). We considered mutants displayed a higher frequency of basal stress or hyperresponsive cells if they had at least >25% of hyper-responsive cells.

To identify genotypes with higher variability, we first calculated the mean and standard deviation of the Upregulated 200 signature for the two wild type cells included in the same experiment. We considered mutants with a standard deviation greater than the mean + 2 SD from the wild type threshold.

## Ranking of signature gene usage

To calculate the number of cells expressing each of the induced and repressed signatures, we used the DotPlot function from Seurat using the entire longitudinal dataset. To this end we plotted the resulting DotPlot$data column to generate the graphs of the percentage of expressing cells for each gene.

**Protein network.** To visualize the protein networks for the mutants selected for each transcriptional phenotype (basal stress, hyperresponsive or highly variable) we used the complete list to generate protein interaction networks using STRING[62]. The network was build using only physical interactions based on experimental data using 0.150 confidence threshold. The resulting networks were exported to Cytoscape[63] for color and layout editing.

## Differential expression

To identify genes with significant expression changes between conditions, we performed a differential expression (DE) analysis using the Seurat package in R. Our analysis focused on comparing two groups within the different datasets.

To generate the expression matrices for differential expression we set two comparisons. First, for basal stressed cells, we generated a comparison matrix based on the "basal_stress" parameter TRUE versus FALSE. Second, to generate clustering only based with stress-induced genes, we made an object that contained the time 0 and time 15 of wild type cells. Then, we generated a comparison matrix between time 0 and time 15 min. For each comparison, we used the FindMarkers function from Seurat to identify differentially expressed genes between groups. We used the build in Wilcoxon rank sum test to determine statistical significance. The genes upregulated above FC ≥ 1.5 (log2 ≥ 0.58) and *pval* < 0.05 were used to perform generate a Seurat object to perform the analysis using only osmostress-induced genes.

## Reporting summary

Further information on research design is available in the Nature Portfolio Reporting Summary linked to this article.

## Data availability

The raw sequencing data and pre-processed data for the longitudinal scRNA-seq profiling and the transcription targeted deletion profiling Gene Expression Omnibus GSE274661. Source data are provided as a Source Data file Source data are provided with this paper.

## Code availability

All the code used in this study is available through Zenodo: 10.5281/zenodo.13731922 [https://doi.org/10.5281/zenodo.13731922]

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

## Acknowledgements

We would like to thank the Functional Genomics Core Facility at IRB Barcelona, the Flow Cytometry Unit of the Scientific and Technological Centers (CCiTUB), the Functional Core Genomics Facility and the Biostatistics Unit at IRB Barcelona for technical support. ICGex NGS platform of the Institut Curie (supported by the grants ANR-10-EQPX-03 (Equipex) and ANR-10-INBS-09-08 (France Génomique Consortium) from the Agence Nationale de la Recherche ("Investissements d'Avenir" program), by the ITMO-Cancer Aviesan (Plan Cancer III) and by the SiRIC-Curie program (SiRIC Grant INCa-DGOS-465 and INCa-DGO-SInserm_12554). We are also grateful to Dr. Pablo Latorre, Dr. Kevin J Verstrepen, Guillem Posas, Dr. Camille Camille Stephan-Otto Attolini for helpful discussions during the initial conceptualization of the project. G.L. and S.P. thank members of the Pelet lab for helpful discussions and Vincent Vincenzetti and Yves Dusserre for technical help. Finally, we would like to thank Aida Fernández for technical support. This work was funded by: PID2021-124723NB-C21/C22 funded by MICIU/AEI /10.13039/501100011033 and ERDF/EU to F.P. and E.dN. Funding from the Ministry of Science, Innovation and Universities through the Centres of Excellence Severo Ochoa Award, and from the CERCA Programme of the Government of Catalonia and the Unidad de Excelencia María de Maeztu, funded by the AEI (CEX2018-000792-M). The Ramon y Cajal Program (Spanish Ministry of Science, RYC2021-033520-I) and La Caixa Junior (LCF/BQ/PR20/11770001) awarded to M.N.R. F.P. and E.deN. are recipients of an ICREA Acadèmia award (Government of Catalonia). The La Caixa Retaining PhD (LCF/BQ/DR21/11880014) to M.Q. S.A. was funded with EMBO Scientific Exchange Grant Number 9387. Work in the Pelet lab was supported by the University of Lausanne and the Swiss National Science Foundation (Grant # 31003A 182431).

## Author contributions

M.N.R., C.S., S.P., A.M., E.dN., F.P. conceptualized the study. M.N.R., G.L., C.S., Y.M., M.Q., M.R., S.A., U.S., A.G. performed the experiments and analyzed the data. M.N.R., E.dN., A.M., S.P., F.P. acquired funding. S.P., E.dN., F.P. supervised the development of the project. M.N.R., S.P., A.M., E.dN., F.P. wrote, reviewed and edited the manuscript.

## Competing interests

The authors declare no competing interests.
