## [Transparent Peer Review file · Nature Communications]

Transcriptional heterogeneity shapes stress-adaptive responses in yeast

Corresponding Author: Dr Francesc Posas

Version 0:

Reviewer comments:

Reviewer #1

(Remarks to the Author)

In this paper, Nadal-Ribelles and colleague report an analysis of budding yeast gene regulation during osmotic stress response at the single cell level using scRNA-seq. This is a thorough and well executed analysis supported by nice validation experiments. I list below a series of comments/concerns that should be addressed prior to publication.

1. Figure 1C-E, and throughout the manuscript: Variance in gene expression is correlated with the mean. Therefore, changes in variability with mean expression are not straightforward to interpret. When commenting on heterogeneity the authors should also report and comment on the Fano factor (variance/mean) calculated from their data. For instance, a Fano factor deviating from 1 reflect deviation from a Poisson process and high Fano values reflect high gene expression noise.
2. Figure 2F: It would be useful to show the number of counts per cell for each cluster in the style of Figure 2G. This would convince the reader that the patterns are not driven by capture efficiency.
3. Text associated with Figure 2G. What does “variability” mean in the context of TFs? Variability between cells or between clusters?
4. It is good to see that the hyper-responders do not show a difference in number of genes/molecules (S3D-E). However, what are the “normalised” values plotted? I feel filtered raw counts should be reported here.
5. Figure 4E: As far as I can tell both data distributions mostly overlap when taking SD into account. Is the increase in mean of the high pHOR7 cells even significant? Couldn't the larger spread due to a few outliers?
6. Figure 4F: For WT cells, were the top 2% GFP expressors used in the competition assay (unclear from the methods)? This should be as cells with global higher translation levels could have a competitive advantage for instance.
7. Page 10, “The expression of stress-inducible genes is accompanied...”. This sentence needs references.
8. Page 11, “Therefore, our findings support the notion that the intensity...” The data presented are purely correlative, this conclusion sentence is too strong. The data show that repressed genes have less Pol II than induced genes but say nothing about Pol II redistribution or it being a limiting factor.
9. Page 13: “This approach allowed us to identify 33 mutants with a high frequency of basal stressed mutants (Figure 6F) (Extended Data Table 7)”. Shouldn't it read “basal stressed cells”?
10. Figure 6J: Do the findings hold if Fano factor is used instead of standard deviation. If not, what does it mean?
11. Figure 6L: Stress cross-protection is not a new concept and has been studied before. This literature should be acknowledged here.
12. Page 18: “We hypothesized that prestressing variable mutants such as *ada2* with a short and mild osmostress...”. In order to be fully tested, this hypothesis would require transcriptomics data to detect high expressor cells. As the authors do

not provide such data, this should be rephrased or discussed.

13. It is unclear to me how the strains and data reported in Figure 6 compare to those from Nadal-Ribelles, M. et al. Perturbation-driven transcriptional heterogeneity impacts cell fitness. bioRxiv 2024.05.31.596868 (2024) doi:10.1101/2024.05.31.596868.

Is there any overlap?

##

(Remarks on code availability)

Reviewer #2

(Remarks to the Author)

Most studies in yeast measure transcription from entire populations of cells grown in batch culture thus giving a read out that represents the average transcriptional response of that population. This most common type of analysis will miss (or ignore) any variation in transcription response that individual cells in the population might have.

In this paper the authors use state of the art single cell RNA-seq analysis in yeast (*Saccharomyces cerevisiae*) to measure transcription within cells of a population that undergoes a time-course osmotic shock. The aim was to reveal the transcriptional time-course of the osmo-response in each cell of the population. The authors show that there is a wide range of different responses of the 200 or so known osmo-response genes that occurs within the cells of the similarly shocked population. Notably, the authors show that only 25% of the well characterised osmo-response genes are expressed in the majority of cells in the population.

The authors were then able to group cells of the population into cohorts in which the osmo-response genes followed particular transcriptional responses. Interestingly, one sub-population of cells displayed a hyper-response to the osmotic stress which the authors hypothesised might offer these cells a fitness benefit to this stress. In a clever and elegant experiment, the authors were able to isolate this population of cells and, using a competition assay, demonstrated that these cells did indeed have higher fitness under stress conditions.

The general understanding of the osmo-stress response is that the 200 or so osmo-stress response genes are activated in response to the stress to enable the cells to adapt to the harsh environment. However, the authors found some cells in the population exhibited Hog1p-independent up-regulation of osmo-stress response genes even in the absence of stress. Using another technically tricky microscopy approach, the authors confirmed this stress independent up-regulation of osmo-stress gene transcription, and went on to show that the cells showing the stress independent up-regulated osmo-stress gene profile had a stronger response to stress. Most impressively, the authors then isolated the cells showing the stress independent gene up-regulation profile and performed competitive fitness experiments with and without stress. Remarkably, the authors revealed that, although these cells had a fitness cost under conditions of no stress, these same cells had a higher fitness when stressed. Thus, this sub population of cells ensures maximum fitness if and when exposed to stress but at the expense of reduced fitness in the absence of stress; a cellular insurance policy for the population.

The authors then showed plausible data that suggested that a global gene repression response via relocation of Pol II and associated factors was required to ensure full induction of the stress response gene signature.

Finally, the authors used modified bar-coded yeast gene knockouts to selectively screen for genes that might be involved in the osmo-response gene heterogeneity they had previously shown. This screen revealed chromatin remodellers and histone modification (acetylation) machinery as being important in regulating the heterogeneity of the osmo-stress gene response in a cell population.

Overall, I thought this was an ambitious study which employed numerous technically challenging, well-designed, and well executed experiments that delivered high quality data that did support their important conclusions. I felt that the study was successful in (i) identifying the heterogeneous transcription profiles of cells within a population exposed to an osmotic shock and (ii) uncovering the intriguing biological significance of this varied transcriptional output with regards fitness. I feel this original study will be of interest to a wide readership and will serve as an important foundation for others to build on to further investigate, for example, the potential chromatin-mediated mechanisms that might govern the different responses of individual cells within a population to the same set of signals.

Minor comments below:

1. This was a very data dense study using techniques (single-cell seq) that many readers may not be familiar with. The authors could help the reader by giving more detail and information in the figure legends and text regarding some of their data to reassure the reader that they are understanding and following text and results correctly.
2. In the legend to Fig 2F, the authors state that expression is indicated by 'warmer colours'. I do not know what a warmer colour is. A key to show the colour/expression scale should be added. Furthermore, for Fig. 2F, it might be helpful to indicate some of the genes, or groups of genes, that are discussed in the main text that are apparently shown in this heat map.
3. There is also no colour/expression scale for the heat map shown in Fig. 3B. Legend just describes 'warmer colours' ?

4. It was not clear to me what the significance of Fig. 3c was. This figure component was not explicitly described in the text. Thus, to the non single cell (UMAP) expert, this figure component was meaningless.
5. I think Fig. 4D was incorrectly referred to in the results text as Fig. S4D as there was no other reference to Fig. 4D. Furthermore, the data in Fig. 4D was not clearly described in the text. I could see the histone genes highlighted in this figure. However, there was no description of why they were highlighted and what this figure meant.
6. Does the term, 'detected' in the table shown in Fig. 6A mean the same thing as 'profiled' as described in figure legend?
7. In the fig legend to Fig. 6, part 'K' was incorrectly labelled as part 'L'.
8. I am not keen on the term 'histone remodelling'. Although this term could be considered correct, although not preferable, when referring to the mode of action of histone post translational modifying machinery, I feel it should not apply to the mode of action of ATP-dep remodelling complexes.

(Remarks on code availability)

Version 1:

Reviewer comments:

Reviewer #1

(Remarks to the Author)

We would like to thank the authors for their thorough revisions. All our comments have been addressed and we wish them goodluck with publication.

(Remarks on code availability)

Reviewer #2

(Remarks to the Author)

I am happy with the edits/corrections made in response to the initial reviewer's comments.

(Remarks on code availability)

REVIEWER COMMENTS

Reviewer #1 (Remarks to the Author):

In this paper, Nadal-Ribelles and colleague report an analysis of budding yeast gene regulation during osmotic stress response at the single cell level using scRNA-seq. This is a thorough and well executed analysis supported by nice validation experiments. I list below a series of comments/concerns that should be addressed prior to publication.

We would like to thank the reviewer for the constructive and positive feedback of the manuscript. We have addressed the concerns raised by the reviewer which we believe nicely complement and strengthen the revised version of the manuscript.

1. Figure 1C-E, and throughout the manuscript: Variance in gene expression is correlated with the mean. Therefore, changes in variability with mean expression are not straightforward to interpret. When commenting on heterogeneity the authors should also report and comment on the Fano factor (variance/mean) calculated from their data. For instance, a Fano factor deviating from 1 reflect deviation from a Poisson process and high Fano values reflect high gene expression noise.

This is an interesting observation. Originally we used the standard deviation as a mean to show the dispersion in the distribution, but we agree that reporting the Fano factor would be informative as well. As suggested, we have calculated the Fano factor for the signatures in Figure 1C-E. The absolute value for Fano factors is low suggesting the variance in expression is low, however the variability increases upon stress. This is consistent with the previous representation of this figure, wild type cells display higher transcription heterogeneity variability upon 15 min of stress. Following the reviewer's suggestion, we have now included a new Supplemental figure displaying the Fano factor (Figures S1E-F) and described it in the main body of the manuscript.

New Figure S1E-F. Barplot of Fano factors (y axis) for the induced (E), repressed (F) and unresponsive osmoconsensus signatures across all samples.

2. Figure 2F: It would be useful to show the number of counts per cell for each cluster in the style of Figure 2G. This would convince the reader that the patterns are not driven by capture efficiency.

Following the reviewer’s suggestion we have now included the number of counts per cell in the Figure 2F. Of note, the total number of molecules and genes is lower for cluster 1 as well as the number of genes. To further understand this difference, we assessed if the expression of repressed and unresponsive genes were also lower for the cells in this cluster.. Interestingly, the expression of the repressed genes and non-stress responsive are higher for cells in this cluster than the rest of the clusters. The higher expression of basal, highly-expressed genes together with the slightly higher abundance of unresponsive genes indicate that the functionality of the basic transcriptional machinery is similar across clusters yet suggests a deficient repression of highly expressed genes in control conditions for this specific cluster. Therefore, our results suggest that the cells in this cluster 1, likely represent a population unable of reshaping their transcriptome upon osmstress supporting a biological difference rather than a technical artifact.

We have now included the numbers requested by the reviewer in the bottom of the figure 2F and included the comparative data with all the subsets of genes as a new Figure 2G and discussed it in the revised manuscript.

New Figures 2F-G. F) Heatmap of marker genes for subpopulations identified in the wild-type 15-minute dataset. Differential expressed genes (fold-change ≥ 1.5 , adjusted p-value < 0.05) obtained by comparing the control and 15-minute conditions were used to identify subpopulations using for Louvain clustering. Yellow colors indicate higher whereas purple colors indicate expression levels. Subpopulation labels are shown at the top of the heatmap and representative genes and cluster-specific gene names are shown as well as the median number of molecules and genes per cell. G) Expression distribution of the induced (top), repressed (middle) and unresponsive (bottom) signatures across the identified subpopulations in 2G. Dotted line indicates the median expression of the entire population as a reference. Statistical significance (Wilcoxon test) of each cluster against the population is shown above.

3. Text associated with Figure 2G. What does “variability” mean in the context of TFs? Variability between cells or between clusters?

The term “variability” in the original text meant to refer the different degrees of transcription factor activities across subpopulations. The message we were trying to convey was that the expression of *Msn2/4* targets showed different intensities across clusters (from low expression

to high expression), yet activities of osmostress-specific transcription factors like Sko1 displayed less variability. The text referred to figure S2E only and not 2G. We agree with the reviewer than referring to both figures is misleading and thus, we have now modified the text to clarify this point.

4. It is good to see that the hyper-responders do not show a difference in number of genes/molecules (S3D-E). However, what are the “normalised” values plotted? I feel filtered raw counts should be reported here.

We thank the reviewer for the observation. We obtained normalized expression data from the Seurat object after having performed the normalization and pre-processing (e.g. cell cycle regression). Next, we used them used to calculate the number of genes detected and the total number of transcripts per cell. We have now clarified this point in the manuscript. Moreover, as suggested by the reviewer, we have also plotted now the filtered raw counts (nCount_RNA) and genes (nFeature_RNA) to demonstrate that the hyper-responders do not show a difference in genes/molecules. We have included this as new Figure S3D-S3E and modified the figure legend accordingly.

New Figure S3D-S3E. D-E) Distribution of the number of genes (D) and number of molecules (E) for the hyper-responsive (red) and the rest of the population (grey). Points represent cells in the indicated populations. Black line represents the mean abundance while dotter lines the population mean. Statistical significance (Wilcoxon test) is shown between groups.

5. Figure 4E: As far as I can tell both data distributions mostly overlap when taking SD into account. Is the increase in mean of the high pHOR7 cells even significant? Couldn't the larger spread due to a few outliers?

We agree with the reviewer that in Figure 4E there is a large spread in gene expression. To assess the significance in the increase of *HSP12* expression in higher pHOR7 expressing cells, we compared the mean expression of both populations (low and high HOR7 expressing cells) at its maximal expression level. Despite both populations display high variability, the mean of *HSP12* expression is significantly higher for basal *HOR7* expressing cells ($pval = 4.0201e-11$, Welch's t.test). We have now included the statistical analysis in a new Figure S4F and modified the manuscript accordingly

New Figure S4F. Comparison of the pHSP12-PP7-mCherry reporter from Figure 4E. Single cell traces were sorted in two sub-populations based on the level pHOR7-MS2-GFP under basal conditions. For each cell in each subpopulation the maximal HSP12 expression is plotted (points). The expression distribution of each subpopulation is shown (boxplot) and the Welch t.test was performed to compare the mean expression between the low and high cells.

6. Figure 4F: For WT cells, were the top 2% GFP expressors used in the competition assay (unclear from the methods)? This should be as cells with global higher translation levels could have a competitive advantage for instance.

The competition assay for Figure 4F was performed using random sorted GFP cells as a WT control. We have now included an additional control in the competition experiments to confirm that it is the expression of *HOR7* that alters fitness in control and stress conditions. As suggested by the reviewer, we sorted the top 2% population of wild type cells constitutively expressing *SAG1*, a gene that is similar to *HOR7* as its expression is low under basal conditions and regulated by the pheromone pathway. Reassuringly, the top 2% *SAG1* expressing cells did not alter competitive fitness when compared to a wild type strain (grey). This reinforces the idea that the transient expression of *HOR7* leads to a better fitness upon stress but reduced fitness under normal conditions and suggests that the fitness benefit observed is specific to the stress response mediated by *HOR7*, rather than an artifact that provides non-specific competitive advantage. We have now included this control in the revised version of the manuscript (new Figure 3F).

New Figure 4F. Top 2% of pHOR7-mCherry and pSAG1-mCherry cells control conditions or after 1 h 0.4M NaCl were isolated by Flow Cytometry and mixed at a 1:1 ratio with wild random sorted wild-type cells carrying constitutive GFP (10,000 cells/strain). This initial mixture (t0) was then grown in the in the indicated conditions. The fitness of each strain was determined by FACS after 48 h. Bar plot indicates the percentage of each strain. Error bars represent the standard deviation of three independent biological replicates. Statistical significance comparing the mean abundance to the reference timepoint (t0) is shown (paired t.test).

7. Page 10, “The expression of stress-inducible genes is accompanied...”. This sentence needs references.

The thank the reviewer for pointing this out. We have now included the corresponding references.

1. de Nadal, E., Ammerer, G. & Posas, F. Controlling gene expression in response to stress. *Nat Rev Genet* 12, 833–45 (2011).
3. Gasch, a P. et al. Genomic expression programs in the response of yeast cells to environmental changes. *Mol Biol Cell* 11, 4241–57 (2000).
7. Miller, C. et al. Dynamic transcriptome analysis measures rates of mRNA synthesis and decay in yeast. *Mol Syst Biol* 7, 458 (2011).
27. Ho, Y. H., Shishkova, E., Hose, J., Coon, J. J. & Gasch, A. P. Decoupling Yeast Cell Division and Stress Defense Implicates mRNA Repression in Translational Reallocation during Stress. *Current Biology* 28, 2673-2680.e4 (2018).

8. Page 11, “Therefore, our findings support the notion that the intensity...” The data presented are purely correlative, this conclusion sentence is too strong. The data show that repressed genes have less Pol II than induced genes but say nothing about Pol II redistribution or it being a limiting factor.

We agree with the reviewer that our initial sentence for this data was not clear enough. The analysis of RNA Pol II was meant to show that the repressed genes display a stronger drop in RNA Pol II, represented by the ratio of reads in NaCl/control conditions, when compared to unresponsive genes. However, it is true that this is a correlative observation. Therefore, we have rephrased the conclusion to clarify this point.

9. Page 13: “This approach allowed us to identify 33 mutants with a high frequency of basal stressed mutants (Figure 6F) (Extended Data Table 7)”. Shouldn’t it read “basal stressed cells”?

We thank the reviewer for the observation. Yes, this should read “basal stressed cells” we have fixed this in the revised manuscript.

10. Figure 6J: Do the findings hold if Fano factor is used instead of standard deviation. If not, what does it mean?

As suggested by the reviewer we have now calculated the Fano factor for the mutants from the Figure 6J. To assess if the findings hold, we have compared the Fano factor and the standard deviation. For both control and stress conditions we observed an excellent correlation; Pearson >0.99 and 0.97 for control and stress respectively, suggesting a consistency between both

metrics. We have now included this information in the revised manuscript and added the Fano factor per each genotype and condition to the Extended Data Table 7.

K

New Figure S6J. Scatter plot shows the comparison between variability metrics of the induced osmoconsensus signature for NaCl. Per each genotype the standard deviation (x axis) and the Fano factor were calculated. Points are colored by density with yellow indicating the highest point density. The Spearman correlation is shown on top.

11. Figure 6L: Stress cross-protection is not a new concept and has been studied before. This literature should be acknowledged here.

Our initial intention was not meant to say that the concept was new, however this is a nice suggestion and we have now included the acknowledgment and cited the literature correspondingly. New references are highlighted in yellow in the text.

We have included the following citations related to stress cross-protection:

38. Świącilo, A. Cross-stress resistance in *Saccharomyces cerevisiae* yeast—new insight into an old phenomenon. doi:10.1007/s12192-016-0667-7.

40. Warringer, J., Ericson, E., Fernandez, L., Nerman, O. & Blomberg, A. High-resolution yeast phenomics resolves different physiological features in the saline response. *Proc Natl Acad Sci U S A* 100, 15724–15729 (2003).

41. Dhar, R., Sägesser, R., Weikert, C. & Wagner, A. Yeast Adapts to a Changing Stressful Environment by Evolving Cross-Protection and Anticipatory Gene Regulation. *Mol Biol Evol* 30, 573–588 (2013).

42. Dragosits, M. & Mattanovich, D. Adaptive laboratory evolution - principles and applications for biotechnology. *Microb Cell Fact* 12, 1–17 (2013).

43. Beaumont, H. J. E., Gallie, J., Kost, C., Ferguson, G. C. & Rainey, P. B. Experimental evolution of bet hedging. *Nature* 2009 462:7269 462, 90–93 (2009).

12. Page 18: “We hypothesized that prestressing variable mutants such as *ada2* with a short and mild osmostress...”. In order to be fully tested, this hypothesis would require transcriptomics data to detect high expressor cells. As the authors do not provide such data, this should be rephrased or discussed.

Our initial hypothesis originated from the scRNA-seq data in which *ada2* has low and high expresser populations (original Figure S6J). From this observation, we hypothesized that a mild and short pre-stress (0.4M NaCl) would generate variability and a fitness advantage. As such, in our initial experiment we observed that pre-stressed *ada2* mutant cells had better fitness when subjected to high osmolarity than those without pre-stress. However, we agree that despite the experiment yielded the expected results, this claim would require an additional transcriptomic profiling to fully support this hypothesis.

To experimentally strengthen our hypothesis, we have now integrated the *HXT5* reporter (pHXT5-UbiM-mCherry-tHXT5) in wild type and *ada2* mutant cells and measured its expression by FACS in response to stress with or without pre-stressing the cells. We followed the same experimental design in which we shortly exposed cells or not (control) to 0.4M NaCl (pre-stressed) and then subject both populations to a second and higher shock (final concentration of 1M NaCl in both cases). We measured expression and compared the ratio of the *HXT5* median expression of the pre-stressed/not pre-stressed populations for the wild type and *ada2* mutant. Interestingly, *ada2* pre-stressed cells displayed a 2.2X increase in *HXT5* expression compared to the 1.3X increase observed in wild type cells. Albeit this is only a single-gene reporter, it supports that pre-stressing *ada2* could promote the appearance of high-responder cells. However, we understand that this is only a reporter gene and thus, as suggested by the reviewer, we have included this new data but also rephrased our original statement.

M

New Figure S6M. Expression of destabilized *HXT5* reporter in the indicated strains. Cells were pre-stressed or not in the presence of mild osmostress (0.4M NaCl 30 mins) and then subjected to hyperosmotic stress to the same final NaCl concentration (1M). For each strain the median expression of *HXT5* mCherry signal was extracted. Barplot shows the mean of 3 biological replicates with standard deviation.

13. It is unclear to me how the strains and data reported in Figure 6 compare to those from Nadal-Ribelles, M. et al. Perturbation-driven transcriptional heterogeneity impacts cell fitness. bioRxiv 2024.05.31.596868 (2024) doi:10.1101/2024.05.31.596868. Is there any overlap?

This is a very interesting question. As noted by the reviewer, we have recently performed a genome-scale perturbation experiment in control and osmostress conditions (in Biorxiv at this point, and under revision). Both experiments share a similar experimental design although there

are slight technical differences in the library and the scRNA-seq platform. For the large-scale study, as we needed to profile a massive number of cells, we used a microwell based platform (Singleron Biotechnologies) while in this study we used droplet based technology (10X genomics). Therefore, inevitably the data preprocessing is different (CeleScope for Singleron and CellRanger for 10X Genomics). Because the genome-scale dataset is under review we have included this comparison as a rebuttal figure.

As suggested, we have performed a global comparison by reprocessing the overlapping strains from both studies. First, we clustered all genotypes. In agreement with this study, cells predominantly cluster by stress and not by genotype (Rebuttal Figure 1A,1B). This UMAP structure is common to this study as well as our genome-scale analysis.

Besides the UMAP structure, one of the main conclusions from our genome-scale study that relates to this work as well is the presence of cell states with expression of specific gene programs (e.g. daughters, old cells). To this end, we scored the expression of several gene signatures from our genome-scale study into the current dataset. Interestingly, targeted perturbation screen recapitulate several of these states with nearly identical expression of gene modules. For example, both daughter and old cells are defined by the expression of the daughter specific program (DSE-genes and expression of iron-starvation like genes (FIT2, FIT3), as well as other metabolic cell states (e.g. phosphate starvation). This suggests that gene expression patterns underlying cell states are robust and detectable across experiments (Rebuttal Figure 1C, 1D) Among the cell states described in our genome-scale perturbation, we identified basal stressed cells (cluster 14). In this study, the markers defined by that population (HOR7, PNC1, among others) are also upregulated suggesting that we observed the induction of similar genes in this subpopulation of basal stressed cells (Rebuttal Figure 1E).

Despite that the expression of the stress signature is low in the absence of stress and taking into account the differences in the methodology used, we scored the expression of the basal-stress signature reported in this study in the genome-scale dataset. Of note, 21 out of the 46 basal stressed genotypes in this study were shared with the genome-scale dataset that contains a reduced number of cells per genotype. Nonetheless, the mutants that we identified as “basal stressed” in our current dataset showed a tendency, albeit not significant, to accumulate towards cluster 14C (odds ratio >1) suggesting that these mutants in the large scale perturbation screen enrich towards the basal-stressed cluster (Rebuttal Figure 1F)

Furthermore, because the expression of the induced osmoconsensus is higher upon stress, we performed the same analysis. As such, we found a good correlation between the expression of the induced osmoconsensus across both studies, suggesting that both datasets efficiently capture the transcriptome changes upon stress (Rebuttal Figure 1G). Our transcription-targeted perturbation approach has surprisingly revealed a set of negative regulators of osmoresponsive program (hyper-responsive mutants). Indeed, these hyper-responsive mutants in the genome-scale dataset display higher expression reinforcing the existence of a negative regulator network (Rebuttal Figure 1H). Thus, we believe that these results provide an overall view on the concordance between both datasets in capturing and reproducing gene expression patterns.

Rebuttal Figure 1. A-B) UMAP of overlapping genotypes between transcription targeted perturbation screen using the genome-scale perturbation dataset (control, $n=XX$, $y=YY$). Overlapping genotypes were re-processed as a new dataset, normalized and re-clustered. Cells are colored by condition (A) or by genotype (B). C) Expression of representative cell state marker genes identified in the transcription targeted dataset (this study) as a function of the cell states identified in the genome-scale perturbation dataset. Dot size represents the percentage of expressing cells and color the expression of the indicated gene (y axis). D) Distribution of odds ratios for mutants in cluster 14, highlighting basal-stressed mutants from a genome-wide dataset. The violin plot depicts the enrichment of basal-specific mutants, with the dotted line representing the global genome-wide odds ratio (Odds ratios >1 indicate mutant enrichment). Black line indicates group mean, individual data points and Wilcoxon test are shown. E) Median expression genome-scale dataset (NaCl) (Nadal-Ribelles et al., 2024 Biorxiv) vs. Median expression induced osmoconsensus (NaCl) (Transcription targeted. This study) showing a positive correlation ($r = 0.5$). F) Median expression induced osmoconsensus genome-scale (Nadal-Ribelles et al., 2024 Biorxiv) for Rest and Transcription targeted Hyper-responsive mutants, showing a significant difference (****).

Comparison of the median expression of the induced osmoconsensus for the indicated dataset. Each point shows a mutant with at least 6 cells. The fitted line with the confidence interval is shown as well as the Pearson correlation. F) Violin plot shows the distribution of the median expression of the induced osmosconsensus for mutants identified as hyper-responsive in the targeted perturbation screen. Black line indicates group mean, individual data points and Wilcoxon test are shown.

Reviewer #2 (Remarks to the Author):

Most studies in yeast measure transcription from entire populations of cells grown in batch culture thus giving a read out that represents the average transcriptional response of that population. This most common type of analysis will miss (or ignore) any variation in transcription response that individual cells in the population might have.

In this paper the authors use state of the art single cell RNA-seq analysis in yeast (*Saccharomyces cerevisiae*) to measure transcription within cells of a population that undergoes a time-course osmotic shock. The aim was to reveal the transcriptional time-course of the osmo-response in each cell of the population. The authors show that there is a wide range of different responses of the 200 or so known osmo-response genes that occurs within the cells of the similarly shocked population. Notably, the authors show that only 25% of the well characterised osmo-response genes are expressed in the majority of cells in the population.

The authors were then able to group cells of the population into cohorts in which the osmo-response genes followed particular transcriptional responses. Interestingly, one sub-population of cells displayed a hyper-response to the osmotic stress which the authors hypothesised might offer these cells a fitness benefit to this stress. In a clever and elegant experiment, the authors were able to isolate this population of cells and, using a competition assay, demonstrated that these cells did indeed have higher fitness under stress conditions.

The general understanding of the osmo-stress response is that the 200 or so osmo-stress response genes are activated in response to the stress to enable the cells to adapt to the harsh environment. However, the authors found some cells in the population exhibited Hog1p-independent up-regulation of osmo-stress response genes even in the absence of stress. Using another technically tricky microscopy approach, the authors confirmed this stress independent up-regulation of osmo-stress gene transcription, and went on to show that the cells showing the stress independent up-regulated osmo-stress gene profile had a stronger response to stress. Most impressively, the authors then isolated the cells showing the stress independent gene up-regulation profile and performed competitive fitness experiments with and without stress. Remarkably, the authors revealed that, although these cells had a fitness cost under conditions of no stress, these same cells had a higher fitness when stressed. Thus, this sub population of cells ensures maximum fitness if and when exposed to stress but at the expense of reduced fitness in the absence of stress; a cellular insurance policy for the population.

The authors then showed plausible data that suggested that a global gene repression response via relocation of Pol II and associated factors was required to ensure full induction of the stress response gene signature.

Finally, the authors used modified bar-coded yeast gene knockouts to selectively screen for genes that might be involved in the osmo-response gene heterogeneity they had previously shown. This screen revealed chromatin remodellers and histone modification (acetylation) machinery as being important in regulating the heterogeneity of the osmo-stress gene response in a cell population.

Overall, I thought this was an ambitious study which employed numerous technically challenging, well-designed, and well executed experiments that delivered high quality data that did support their important conclusions. I felt that the study was successful in (i) identifying the heterogenous transcription profiles of cells within a population exposed to an osmotic shock and (ii) uncovering the intriguing biological significance of this varied transcriptional output with regards fitness. I feel this original study will be of interest to a wide readership and will serve as an important foundation for others to build on to further investigate, for example, the potential chromatin-mediated mechanisms that might govern the different responses of individual cells within a population to the same set of signals.

We thank the reviewer for his/her very positive assessment of the manuscript and insightful suggestions which have improved the manuscript.

Minor comments below:

1. This was a very data dense study using techniques (single-cell seq) that many readers may not be familiar with. The authors could help the reader by giving more detail and information in the figure legends and text regarding some of their data to reassure the reader that they are understanding and following text and results correctly.

As suggested by the reviewer, we have now provided more detailed information throughout the figure legends and methods while trying to keep them in reasonable length.

2. In the legend to Fig 2F, the authors state that expression is indicated by ‘warmer colours’. I do not know what a warmer colour is. A key to show the colour/expression scale should be added. Furthermore, for Fig. 2F, it might be helpful to indicate some of the genes, or groups of genes, that are discussed in the main text that are apparently shown in this heat map.

Following the reviewer’s suggestion, we have included a legend to the Figure 2F to indicate the color/expression scale and labeled some of the relevant genes discussed in the text. Additionally we have included the median molecules/genes per cell per each cluster (see reviewer 1 question 2).

New Figure 2F. Heatmap of marker genes for subpopulations identified in the wild-type 15-minute dataset. Differential expressed genes (fold-change ≥ 1.5 , adjusted p-value < 0.05) obtained by comparing the control and 15-minute conditions were used to identify subpopulations using for Louvain clustering. Yellow colors indicate higher whereas purple colors indicate expression levels. Subpopulation labels are shown at the top of the heatmap and representative genes and cluster-specific gene names are shown as well as the median number of molecules and genes per cell.

3. There is also no colour/expression scale for the heat map shown in Fig. 3B. Legend just describes 'warmer colours' ?

Following the reviewer's suggestion, we have now added the figure legend describing the scale of the colors to the new Figure 3B.

4. It was not clear to me what the significance of Fig. 3c was. This figure component was not explicitly described in the text. Thus, to the non single cell (UMAP) expert, this figure component was meaningless.

We agree with the reviewer that our initial reference to Figure 3C could not be intuitive to all the readers. This figure was intended to show the cluster-specific expression of the clusters defined in Figure 3A and 3B. We have now improved the description in the main text and figure legend to clarify this point.

5. I think Fig. 4D was incorrectly referred to in the results text as Fig. S4D as there was no other reference to Fig. 4D. Furthermore, the data in Fig. 4D was not clearly described in the text. I could see the histone genes highlighted in this figure. However, there was no description of why they were highlighted and what this figure meant.

We apologize for the mistake. The reviewer is absolutely correct. We oversimplified the explanation regarding the differential expression of basal-stressed cells. Indeed, Figure 4D represents the results of the differential expression between basal stress cells and the rest of the

population. The volcano plot highlights both top upregulated and downregulated genes highlighted in orange and blue respectively. While we focused in the upregulated genes, histone genes are significantly downregulated perhaps suggesting basal expression of these stress genes is less frequent in S phase. We have now improved the explanation of this panel in the revised manuscript.

6. Does the term, 'detected' in the table shown in Fig. 6A mean the same thing as 'profiled' as described in figure legend?

Yes, the reviewer is right. We have now changed the "detected" in Figure 6A to "profiled" to be consistent.

7. In the fig legend to Fig. 6, part 'K' was incorrectly labelled as part 'L'.

Thank you for pointing this out. We have fixed the labeling as suggested.

8. I am not keen on the term 'histone remodelling'. Although this term could be considered correct, although not preferable, when referring to the mode of action of histone post translational modifying machinery, I feel it should not apply to the mode of action of ATP-dep remodelling complexes.

The reviewer is correct in pointing out that the term 'histone remodeling' might not be the most appropriate and should not apply for ATP remodeling complexes. We have now replaced histone remodeling for ATP-dependent remodeling complex.